# U and Th zonation in apatite observed by synchrotron X–ray fluorescence tomography and implications for the (U–Th)/He system

Francis J. Sousa[1], Stephen E. Cox[2], E. Troy Rasbury[3], Sidney R. Hemming[2], Antonio Lanzirotti[4], Matthew Newville[4]

[1]College of Earth, Ocean, and Atmospheric Sciences, Oregon State University, Corvallis, OR, USA
[2]Lamont–Doherty Earth Observatory, Columbia University, Palisades, NY, USA
[3]Department of Geosciences, Stony Brook University, Stony Brook, NY, USA
[4]Center for Advanced Radiation Sources, The University of Chicago, Chicago, IL, USA

*Correspondence to*: Francis J. Sousa (francis.sousa@oregonstate.edu)

**Abstract**

Synchrotron X–ray fluorescence microtomography can non-destructively image the three-dimensional distribution of several trace elements in whole apatite crystals at the resolution of one $\mu m^3$. This allows for precise determination of the physical geometry of a crystal and the quantification of the relative abundance of the radioactive parent nuclides uranium and thorium, with high fidelity. We use this data to develop a more precise alpha ejection correction for (U–Th)/He thermochronology and high-resolution models of apatite crystals that are the foundation for a new generation of three-dimensional diffusion modeling. The application of synchrotron radiation to non-destructive imaging of minerals used for geochronology sheds light on causes of longstanding unresolved problems in the field that are rooted in previously unmeasurable parent nuclide zonation, especially the pervasive overdispersion of single crystal ages.

**Short Summary**

We have discovered a new way of measuring the three-dimensional distribution of radioactive elements in individual crystals by shining a very bright light on apatite crystals at the Advanced Photon Source at Argonne National Laboratory. This allows us to learn about the rates and timing of geologic processes, and to help resolve problems that previously were unsolvable because we had no way to make this type of measurement.

## 1 Introduction

Compositional zoning is common in apatite and zircon, and its impact on geochronology has been a longstanding concern in the field (e.g. (Hughes and Rakovan, 2015; Hanchar and Miller, 1993; Farley et al., 2011; Ault and Flowers, 2012; Hourigan et al., 2005; Boyce and Hodges, 2005; Pickering et al., 2020; Paul et al., 2019; Flowers and Kelley, 2011; Johnstone et al., 2013; Chew and Spikings, 2015; Fox et al., 2017)). Zoning is particularly important when it affects the distribution of radioactive nuclides and their decay products in crystals targeted for measuring geologic time. Consequently, (U–Th)/He

thermochronology will benefit from a method of non-destructively measuring the three-dimensional (3D) distribution of uranium and thorium.

In the (U–Th)/He system, the combination of helium ingrowth (by alpha decay) and diffusive loss (controlled thermally)—

yields a thermochronologic age. Because of the long stopping distance of alpha daughter products (roughly 20 µm for apatite; (Ketcham et al., 2011; Farley et al., 1996)), some fraction of the alpha decays near a crystal face will be ejected from the crystal upon decay. This is accounted for with a correction that estimates the retained fraction of total alphas (or $F_T$). This is commonly carried out following the framework established by Farley et al. (1996), using a surface area to volume ratio derived from a limited number of crystal dimensions typically measured with a microscope. Several updates to this framework have been

introduced using geometric reconstructions from optical images (Cooperdock et al., 2019), laboratory-based X–ray fluorescence microtomography (Hanna and Ketcham, 2017), and a Monte Carlo statistical approach (Gautheron et al., 2012). While these approaches have improved crystal geometry reconstruction, they provide no information about parent isotope zonation. Published approaches to measuring zonation require sacrificing one or more crystals from a population to a destructive mounting and polishing procedure, e.g. Farley et al. (2011). A model is then used to apply a measured zonation

pattern (usually in just one internal slice of one crystal, for practical reasons) to other crystals. Another less commonly used technique is to extrapolate data from a minimally destructive crystal surface laser ablation map on one external face to the 3D volume of a crystal, e.g., Hourigan et al. (2005).

Given the difficulties of accounting for zonation, the assumption of parent nuclide homogeneity remains ubiquitous in many

applications of (U–Th)/He thermochronology (e.g. (Gavillot et al., 2018; Lamont et al., 2023; Sousa, 2019; Sousa et al., 2017; Sousa et al., 2016)). Pervasive inter-crystal age dispersion of circa 10 to 15% is significantly larger than commonly reported individual age uncertainties of circa 3 to 4%. This has led to several published efforts exploring the potential impact of parent isotope zonation on single crystal (U–Th)/He ages. Hourigan et al. (2005) demonstrate that the assumption of homogeneity in parent nuclide distribution can regularly lead to errors as large as 30% due to the impact of zonation on the alpha ejection

correction. They present a model for accounting for the impact of parent nuclide zonation on alpha ejection using an $F_{ZAC}$ correction factor (zonation dependent bulk retentivity) rather than the commonly used $F_T$ parameter (Farley et al., 1996). However, the method Hourigan et al. (2005) suggest for measuring zonation is insufficient to solve the problem. It is neither three-dimensionally complete—and therefore requires extrapolation of data from one crystal face to the entirety of the crystal—nor is it non-destructive—polishing and laser ablation both destroy part of the crystal, which must be accounted for

in subsequent whole crystal measurements.

Using a modeling approach, on the other hand, Ault and Flowers (2012) explore the impact of parent isotope zonation on distributions of (U–Th)/He ages, concluding that for many geologic samples, the impacts of zonation are limited to the same magnitude as inter-crystal age dispersion, approximately 15%. While this conclusion is well demonstrated by Ault and Flowers

(2012), their suggestion that assuming homogeneity in geologic samples is "practical and sensible" in the context of ~ 15% inter-crystal dispersion is not justified. Specifically, the question of whether any significant amount of this dispersion may be directly caused by zonation is left unanswered. **If dispersion is caused by zonation, could it be explained–and eliminated– if better techniques for imaging U and Th zonation are developed?** A third study of the impact of U and Th zonation on (U–Th)/He ages by Farley et al. (2011) concludes that zonation is an important source of error in age calculations for (U–

Th)/He ages and $^4$He/$^3$He spectra and suggests that further mapping of parent isotope distributions in apatite "seems worthwhile". Such further mapping, with the twist of being non-destructive, is the immediate pursuit of this work.

## 2 Methods

### 2.1 Analytical Summary

We have acquired data during five separate sessions at the Advanced Photon Source (APS) at Argonne National Laboratory
from February 2020 to March 2023, ranging in length from three to four days. The first session was primarily spent investigating the manipulation of apatite crystals with the motorized goniometer and crystal mount system, apatite microtomography data acquisition, processing, and visualization, and exploring the analytical parameters required for different crystal imaging resolutions, compositional uncertainties, and crystal sizes.

We report here data from four geologic samples. Samples 03PH307A, 69APR32A, and MGB5 are from a suite of bedrock samples from a study of exhumation related to the Fairweather fault in southeast Alaska (Mcaleer et al., 2009). Sample 03PH307A is from a granodiorite collected at 243 m above sea level (59.4378° N, 138.2492° W). It has a mean corrected apatite (U–Th)/He age of $2.34 \pm 0.33$ Ma based on two multicrystal aliquots totaling 16 individual crystals. 69APR32A is from a granodiorite collected at 2499 m above sea level (60.3017° N, 139.6017° W) with a mean corrected apatite (U–Th)/He age
of $1.90 \pm 0.14$ Ma based on two multicrystal aliquots totaling ten individual crystals. MGB5 is from an altered granitoid at 128 meters above sea level (58.8941° N, 136.9230° W) with a mean apatite (U–Th)/He age of $5.90$ Ma $\pm 0.42$ Ma based on four multicrystal aliquots comprising a total of 38 crystals. For the (U–Th)/He data reported by Mcaleer et al. (2009), alpha ejection corrections were made using the methodology presented by Farley et al. (1996). $F_T$ was calculated using an aliquot average based on the measured dimensions of the crystals that comprise the aliquot, as was common practice at the time the samples
were analyzed (e.g. (House et al., 1997)). Sample AP is from sample PRR–15064, a granite from the Byrd Glacier region in the Transantarctic Mountains (80.058° S, 160.298° E; IGSN: PRR015064) that we obtained from the Polar Rock Repository. PRR-15064 does not have published (U–Th)/He data.

Data reported here include at least one virtual slice of tomography data from eight crystals: seven from the Alaska samples
(three from 03PH307A, two from 69APR32A, two from MGB5) and one from the Antarctic sample AP. For this study crystals were selected manually using stereoscopic microscopy under plane- and cross-polarized light at the Lamont–Doherty Earth

Observatory (LDEO). The primary consideration in crystal selection was fully intact terminations and absence of visible inclusions, which is similar to the criteria used in most (U–Th)/He studies (Farley, 2002; Ehlers and Farley, 2003; Reiners et al., 2005; Cooperdock et al., 2019). We experimented with different resolutions and different crystal sizes within the constraints provided by our allocated time, exploring the trade-offs between voxel resolution, crystal size, and time required to image a crystal. As we initially explored the efficacy of synchrotron X–ray fluorescence (XRF) microtomography to image the U and Th distribution in apatite, we prioritized collecting data on multiple crystals over whole crystal 3D imaging. After this initial phase, the primary analytical goal was to measure at least one crystal at high resolution to demonstrate the method's capability to non-destructively document parent nuclide zonation in apatite and create a 1 $\mu m^3$ 3D crystal model to determine $F_T$.

## 2.2 Sample MGB5

We targeted sample MGB5 for high-resolution full crystal imaging. This involved collecting 148 virtual slices with a 1 $\mu m$ step size between virtual slices that cover MGB5–1 and 108 virtual slices with a 1 $\mu m$ step size between virtual slices that cover MGB5–2. During the analysis of MGB5–1, data from one of the slices was lost due to a beam issue at the APS. As a result, we do not use the data from this crystal in our further analyses. A movie of the 3D uranium reconstruction of this grain is available in the video supplement (Sousa et al., 2024a). For MGB5–2, we achieved the highest 3D resolution (1 $\mu m^3$) for the entirety of the crystal. We use this data for 3D reconstructions and alpha ejection calculations (Figure 1). MGB5–2 has dimensions of approximately 60 $\mu m$ in diameter by 120 $\mu m$ in length (Figure 1). While this crystal would be near the lower size limit of a crystal typically selected for an apatite (U–Th)/He study, its small size facilitated full 3D tomographic imaging at high resolution during the analytical time allotted for our experiments at the APS.

We chose this sample due to the ready availability of separates, the existence of published (U–Th)/He data, and the fact that it is a plutonic bedrock sample that typifies common targets for apatite (U–Th)/He studies. While the interpretation of the sample thermal history has important tectonic implications (Lease et al., 2021; Mcaleer et al., 2009; Fletcher and Freymueller, 2003; Witter et al., 2021), the tectonics and exhumation history of the Fairweather Fault in southeast Alaska are not the focus of this study. This study demonstrates the application of synchrotron XRF microtomography for non-destructively imaging the distribution of U and Th in individual crystals of apatite in the context of a real sample plucked from the existing literature.

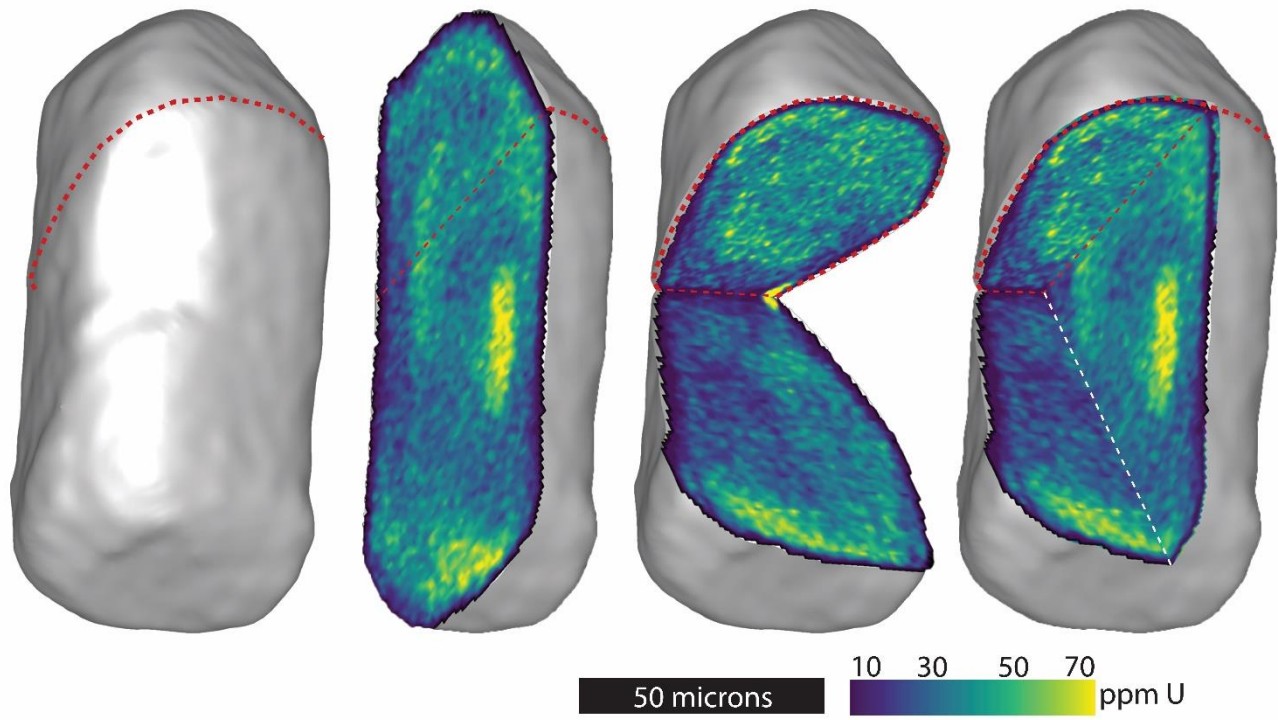

**Figure 1.** Four visualizations of the 3D reconstruction based on synchrotron XRF microtomography data from the MGB5–2 crystal. On the
left is the 3D model of the outside of the crystal. All four frames show views of the crystal from the same perspective with progressively
increasing numbers of cutaway planes (one, two, and three increasing from left to right) and show the internal uranium distribution as
measured with XRF microtomography. The three cutaway planes in the right-hand frame are in the exact orientation and location as the
planes shown separately in the middle two frames. Red dashed lines show the physical orientation of the virtual tomography slices from the
synchrotron XRF analysis. The white dashed line highlights the intersection of cutaway planes. The color scale shows the calculated U
concentration in ppm.

## 2.3 Exploratory Phase Samples

The first virtual slices we measured were from two different locations on 03PH307A–1. Both slices revealed a uranium
zonation pattern with a highly enriched rim (Figure 2A and 2B). Next, we collected one virtual slice from a second crystal,
03PH307A–2 (Figure 2C). This data also showed a partial high uranium rim but was plagued by other high uranium hot spots
that created visualization artifacts. Next, we imaged crystal 03PH307A–3 (~70 µm diameter by ~110 µm length) with 57
virtual slices with 2 µm gaps between slices.

We next collected a single slice from two crystals from 69APR32A (69APR32A–1 and 69APR32A–2). The 69APR32A–1
slice was very small near the crystal's termination but showed a fairly homogenous uranium distribution (Figure 2D). The
69APR32A-2 slice was larger but had several high uranium inclusions that were likely small zircons (Figure 2E). Because of
the homogeneity of uranium distribution of 69APR32A–1, we chose to image the entire crystal (~50 µm diameter by ~90 µm
length) with 38 additional slices and 3 µm gaps between slices crystals. The two 3D reconstructions with inter-slice gaps

(03PH307A–3 and 69APR32A–1) show artifacts in their 3D reconstruction due to the gaps between virtual slices. Movies showing the 3D uranium distribution are available in the video supplement (Sousa et al., 2024a). Several of these early

tomography results imaged likely zircon inclusions. These small inclusions are difficult to see clearly in a flattened two-dimensional (2D) image due to the wide spacing of the virtual slices, but they are visible in the movies included in the video supplement (Sousa et al., 2024a). We include here data from crystal AP1 that clearly shows a zircon inclusion. We discovered this inclusion during the analysis of this crystal at 2 µm resolution. After completing the full crystal imaging at 2 µm, we returned to the zircon inclusion and acquired a single slice through it at 1 µm resolution (1 µm x step and 1° rotation resolution).

This slice demonstrates the capability of the methodology to image and quantify such zircon inclusions in apatite (Figure 2F).

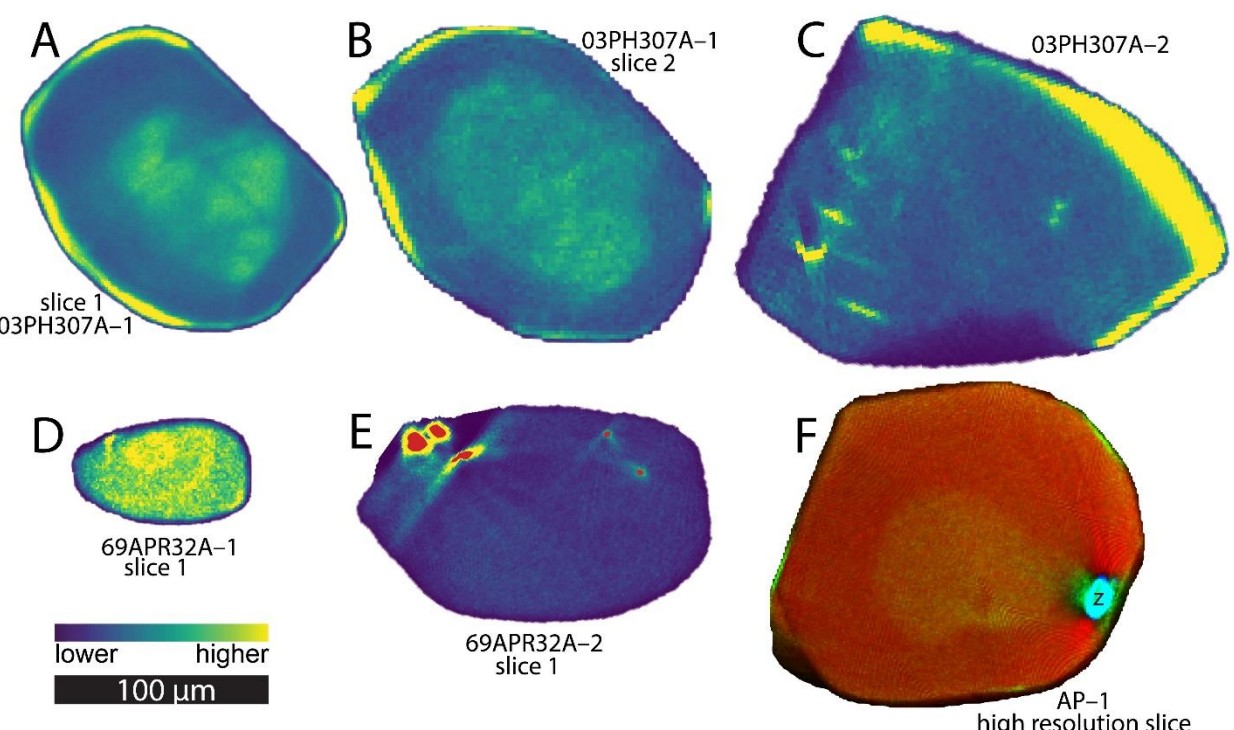

**Figure 2.** Tomographic virtual slices. **A.** and **B.** show two slices from different locations in the same crystal, 03PH307A–1, both showing high uranium enriched rim. Tomography settings for slice 1 (**A**) are 1 µm x steps and 1° angular steps, resulting in 1 µm² pixels. Tomographic
settings for slice 2 (**B**) are 2 µm x steps and 2° angular steps, resulting in 2 µm² pixels. **C.** Virtual tomographic slice from crystal 03PH307A-2. Tomography settings for this slice are 2 µm x steps and 1.5° angular steps. This is the only virtual slice analyzed for this crystal. **D.** Slice 1 from crystal 69APR32A-1. This virtual slice was smaller than the others we analyzed but showed relatively homogeneous uranium distribution, which led us to image the crystal in 3D, as described in the text. Tomography settings for this slice are 1 µm x steps and 1.5° angular steps. **E.** Slice 1 from crystal 69APR32A-2. Several putative zircon inclusions are highlighted in the figure in red color.
Tomography settings for this slice are 1 µm x steps and 1.5° angular steps. This is the only virtual tomographic slice from this crystal. **A-E** use the color scale shown on the figure, which is relative with roughly one order of magnitude variation within each of the virtual slices. **F.** RGB composite image of one slice of crystal AP–1. The location of a zircon inclusion is labeled "z". Red = strontium. Green = uranium. Blue = zirconium. The length scale bar is the same for the entire figure. Underlying XRF data is available in Sousa et al. (2024b). See text of Section 2.5 for explanation of how computed tomography is used to acquire concentration data shown in virtual slices.

## 2.4 Synchrotron XRF Microtomography Methods

This work used the GSECARS X–ray microprobe at Sector 13 (13–ID–E) at the APS. The 13–ID–E station uses microfocusing mirrors in a Kirkpatrick–Baez (KB) geometry to produce a focused X–ray beam of 1 (V) x 2 (H) µm (Sutton et al., 2017), which was the focused spot size used in these experiments. Beamline 13–ID–E uses a 3.6 cm period undulator source and a cryogenically-cooled double-crystal monochromator that allow for focused monochromatic beams with energies between 2.4–28 keV. For the analyses shown here, an incident beam energy of 18.5 keV was used with an incident flux to the sample ($I_0$) of $\sim 8 \times 10^{11}$ photons per s. A Canberra/Mirion SX7, 7-element silicon drift diode detector sitting at 90° to the incident beam is used to measure the emitted XRF from the sample. Pulse processing was done with a high-speed Quantum Xpress3 digital spectrometer system.

For each analysis, an apatite crystal was mounted with epoxy on the end of a 100 µm diameter clean silica fiber. The fiber mounted sample was then placed in modeling clay in a motorized goniometer assembly, which allows for precise centering of the sample relative to the rotation axis and the focused incident X–ray beam (Figure 3). Because whole, unsectioned apatite crystals are used in this experiment, the attenuation of fluorescence X–rays through the crystal diameter that are detected must be considered. These were found to be unacceptable for elements like Ca and Fe but acceptable for the primary target elements U and Th. The calculated transmitted XRF fraction for fluoro-apatite $Ca_5(PO_4)_3F$ with a density of 3.2 g/cm$^3$ and thickness of $\sim 100$ µm is 0.00069 Ca Kα, 0.0062 Fe Kα, 0.49 Th Lα, 0.54 U Lα, 0.57 Sr Kα, 0.62 Y Kα, 0.66 Zr Kα.l

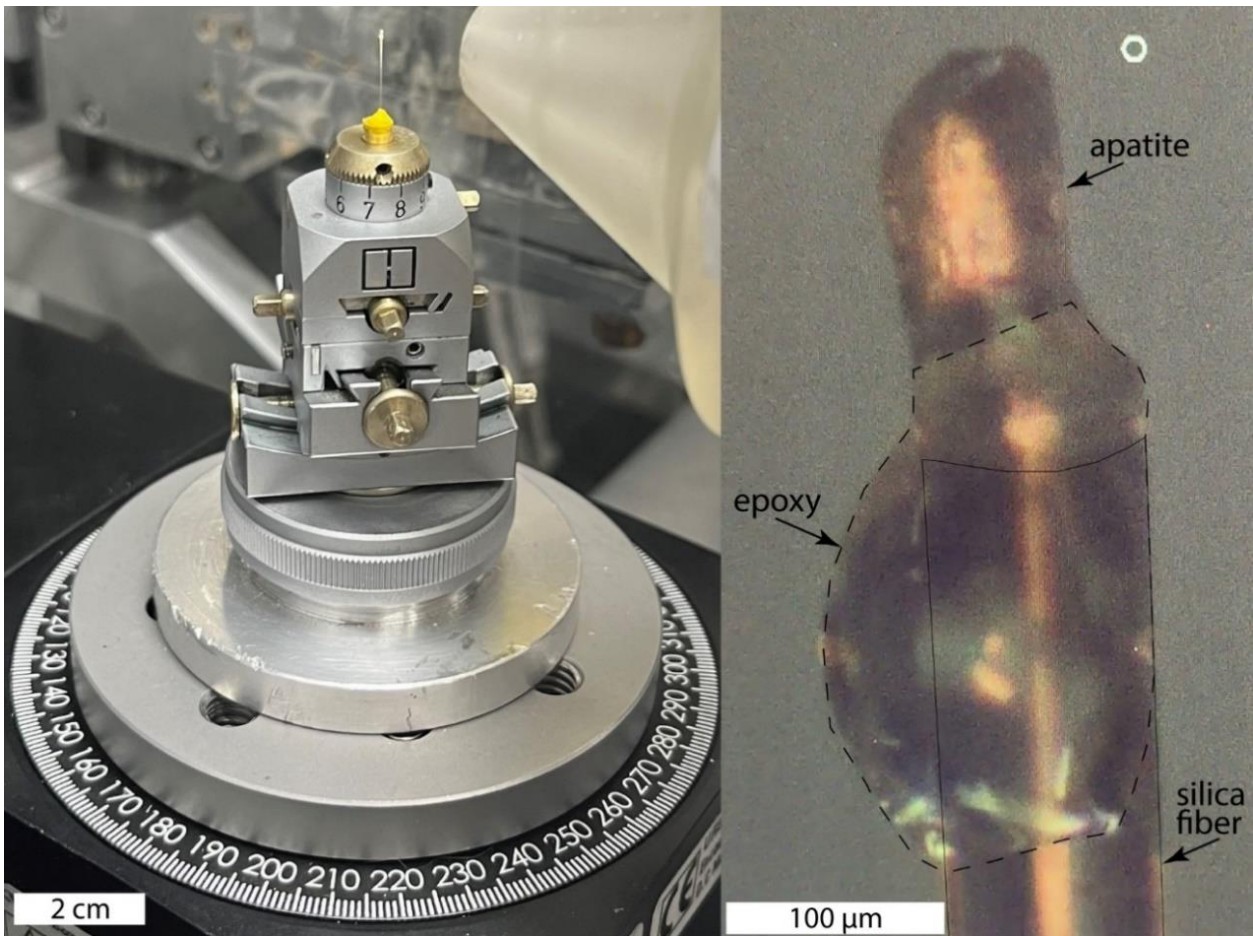

**Figure 3.** Images of rotation stage and motorized goniometer assembly holding crystal mount in epoxy on silica fiber. The left frame shows a crystal set in epoxy on the tip of a 100 µm silica fiber with its base embedded in yellow modeling clay mounted inside a brass pin held in place atop a goniometer setup with a set screw. The right frame is a microscopic view of the end of the silica fiber where the crystal is mounted in epoxy. Silica fiber is annotated in a thin solid black line. Epoxy bleb is outlined in a thin dashed black line.

XRF microtomography data consist of energy dispersive spectra and a measure of X–ray incident and transmitted flux, collected as a series of 2D virtual slices. Crystals are scanned bi-directionally through the focused X–ray beam, using continuous horizontal translation as the fast axis (x, for 03PH307A–1, slice 1, the x step size is 1 µm) and horizontal rotation as the slow axis ($\omega$, for 03PH307A–1, slice 1, we use 1° angular increments over 360° total rotation). These x-$\omega$ maps or *sinograms* are the initial data that is used to construct a virtual slice. For 03PH307A-1, slice 1, the sinogram consists of 361 rows, corresponding to each of the 1° incremented $\omega$ positions from 0° – 360° (Figure 4). The time required for data acquisition varies with crystal dimensions and chosen tomography parameters. For analyses reported here the time was typically on the order of 5-15 minutes per slice, which roughly corresponds to imaging of one high resolution full crystal in a day of beamtime.

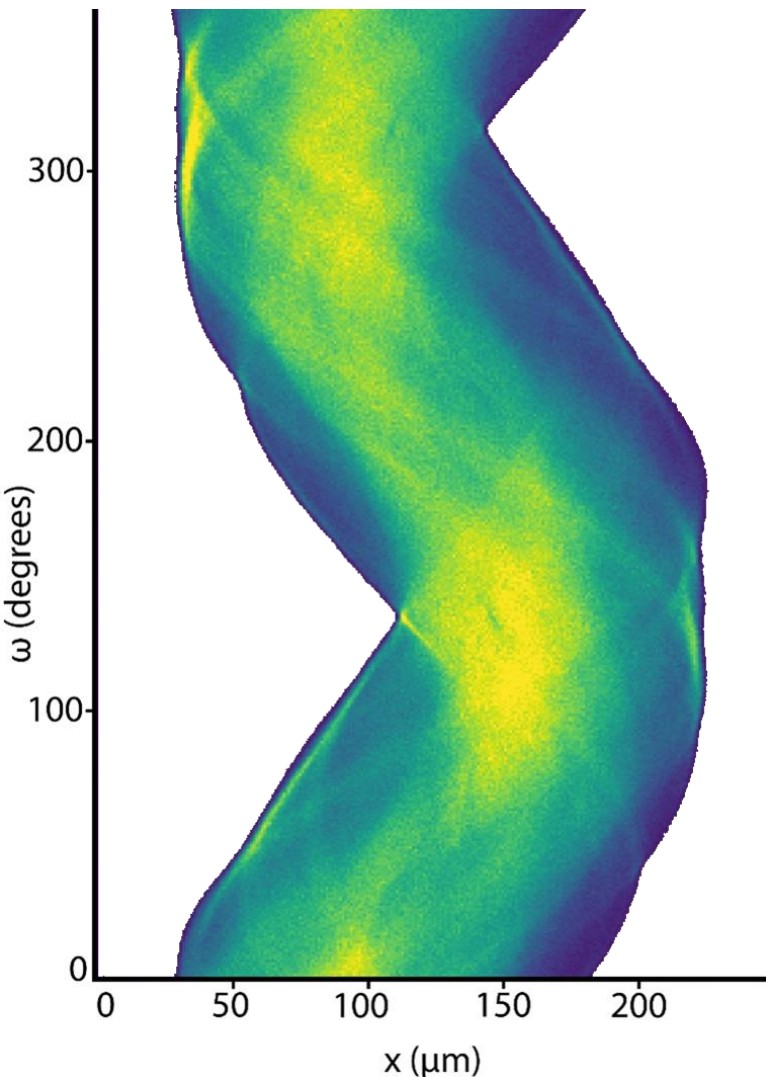

**Figure 4.** Example sinogram from 03PH307A–1, slice 1, as shown in Figure 2A. Color shows the distribution of uranium, using the same relative scale as Figure 2. The horizontal axis shows x steps for each line scan in µm. The vertical axis shows rotational steps, each of which is 1 degree off from the previous. There are 361 rows representing the full rotation from positions 0° – 360°.

## 2.5 Data Reduction and Tomographic Reconstruction Methods

XRF data consist of one-dimensional arrays of full energy dispersive spectra (intensities in 4096 energy channels at 10 eV steps) for each x and ω position (e.g. each pixel in Figure 4), giving a 3D array of (x, ω, E) for fluorescence energy E. These fluorescence intensities are summed over the 7 detector elements and divided by the sampled incident intensity at each (x, ω) position. These (x, ω, E) data are then converted to (x, ω) sinograms of approximate elemental abundances using the X–ray Larch software package (Newville, 2013). These elemental abundances are concentrations in units of ppm as locally determined by analysis of XRF data, with one calibration scaling parameter to match the total abundances of these trace

elements in similar apatite grains. For this conversion, we perform an initial least-squares fit to the total XRF data for the entire sinogram (summed over x and ω). This uses tabulated values for absorption factors, fluorescence yields, air-path to detector, and detector properties while adjusting the detector efficiency and energy broadening terms to determine an overall scaling factor between abundance for each element of interest and the full intensity of its fluorescence. These elemental scaling factors are then calibrated to give reasonable total abundances and used to produce sinograms of abundances for each of the elements of interest (Sr, Y, Th, U). For more details about XRF data processing see Appendix A.

Importantly, this analysis does not account for attenuation of the X–ray fluorescence by the sample itself. This attenuation varies systematically with the energy of the main fluorescence line. Because the X–ray path through the sample to the detector varies for each x and ω value, trying to account for this effect accurately would be difficult to do with sufficient accuracy. The attenuation of fluorescence intensities for the elements of interest here varies by a factor of about 1.5. Thus, while the relative variations of abundance for each of the elements have high fidelity, the absolute abundances reported here for each element can have uncertainties of factors of 3 and should best be scaled using whole crystal elemental concentrations typically measured independently during geochronology applications (e.g. by ICPMS). For the purposes of this study, the critical input for constraining alpha ejection is the relative abundances of U and Th throughout the crystal. Because relative abundances for the elements of interest have high precision, applying a self-absorption correction to further constrain the absolute abundance of elements to a higher precision than we present here is not a primary concern. Appendix B shows the results from a first order self-absorption correction, demonstrating the minor impact it has on the U and Th data presented here. Self-absorption is much more significant for elements with lower energy fluorescence lines such as Ca, but applying a self-absorption correction as in Appendix B does successfully calibrate the total abundance of Ca.

Tomographic reconstructions of the elemental sinograms were done using the TomoPy software package (Gürsoy et al., 2014; Gürsoy et al., 2015). A maximum-likelihood expectation maximization algorithm (MLEM) was used as the iterative reconstruction method (see Gürsoy et al. (2014)), typically using 20 iterations per sinogram. The reconstructed virtual slices of abundance were then saved as a sequence of TIFF images, one for each virtual slice. These TIFF stacks were then used to construct a 3D segmentation of the physical crystal.

**2.6 Building 3D Crystal Models**

Starting with the tomographic outputs of TIFF stacks representing the abundance of U, Th, and Sr, we process the data for crystal MGB5-2 utilizing the workflow explained below. The data processing is completed in two software packages. The first is 3DSlicer for volume manipulation and segmentation construction (version 5.2.1 r31317; Fedorov et al. (2012)). In this context, segmentation is the process by which 3D objects are constructed from stacks of images. For clarity, we point out that while the process is called "segmentation," each of these objects is also individually referred to as a "segmentation". For analysis and alpha ejection modeling we use a Matlab script available in Sousa et al. (2024b).

The first step is to use 3DSlicer to build a segmentation from the stack of tomography images that represents the physical boundary of the crystal of concern. Table 1 defines scientific terms important to understanding the next set of processing steps.


| TERM | DEFINITION |
|---|---|
| voxel | each of an array of cubic elements that collectively constitute a representation of a 3D space, each representing a unit of volume. For the 3D data from MGB5–2, voxels are 1 µm x 1 µm x 1 µm in dimension |
| volume | stores a 3D array of voxels in a rectilinear grid |
| segmentation | a procedure to delineate a 3D region or the resultant 3D model of that region. In this work, segmentation represents both the 3D model of the outside of the crystal and the process of defining that shape |
| labelmap | a 3D scalar volume where each voxel is a number indicating the type of material at that location. In this work, a labelmap has a value of 1 for a voxel inside the crystal and a value of 0 for a voxel outside the crystal |
| $F_T$ | the fraction of total alpha decays retained within the physical boundaries of a particular crystal |
| Helium accumulation array | a 3D array of the same size as the input arrays of U and Th abundance that starts with zero in all voxels. It is used to track the accumulation of alpha particles throughout the iterative decay modeling |
| Helium retained array | Created by multiplying the helium accumulation array by the labelmap, which removes all accumulated He from voxels outside the physical crystal boundary |

**Table 1.** Defining scientific terms related to computational processing of 3D tomography data and modeling alpha decays. While some of these terms are commonly understood in a scientific context, their role in the computational processing described in the text varies from common scientific usage sufficiently to require clear definition. For example, "volume" here does not refer to the space occupied by a physical object, but the computational model of that space stored in a rectilinear grid.


We start by importing the TIFF stacks into 3DSlicer as volumes, and then create a segmentation that represents the physical extent of the crystal. Calcium suffers from strong X–ray self-absorption. Because strontium follows calcium and is minimally impacted by self-absorption, we use it as a proxy for apatite to define the volume of the crystal. The process of creating the segmentation involves both automated processing (thresholding and smoothing) and manual tools checked at each tomographic
slice (scissors and paintbrush) to accurately define the 3D segmentation as the real shape of the crystal. Next, the 3D segmentation representing the physical crystal geometry is exported from 3DSlicer as both a 3D object for visualization purposes (e.g. for 3D printing, see 3D object named 'MGB5-2.ply' in Sousa et al. (2024b)), as well as a labelmap volume file. The labelmap is a 3D array that tracks the (x, y, z) coordinates of the crystal volume using a value of 1 for a voxel within the crystal and 0 for a voxel outside the crystal. This labelmap is imported into Matlab, along with the 3D arrays of U and Th, and

other elemental data output from the tomography processing. In Matlab, the labelmap is first used to clip the extent of the elemental TIFF stack data so that the further processing steps are limited to the physical crystal. This is accomplished by multiplying the labelmap array by each elemental array. This allows for multiple visualizations of elemental data in different orientations using the 3DSlicer platform (e.g. Figure 5). The MGB5-2 3D volume files for U, Th, Sr, and Y are published in Sousa et al. (2024b).


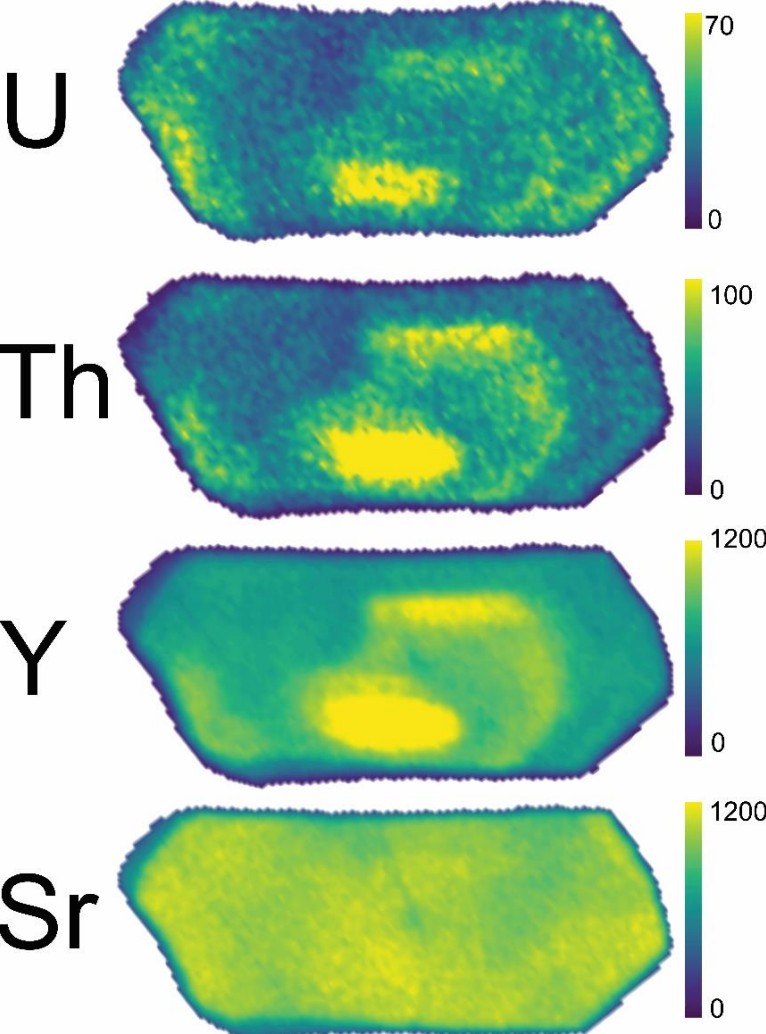

**Figure 5.** Internal slice of MGB5–2 XRF microtomography data for U, Th, Y, and Sr. The color scale shows semiquantitatively calibrated elemental concentration in terms of ppm. All four frames are from the same perspective.

### 2.7 (U–Th)/He Alpha Decay Modeling and Spatial Tracking

We use the U and Th elemental concentration data for the MGB5–2 crystal as the basis for a 3D model quantifying the $F_T$ value. This is accomplished using an alpha decay model that directly tracks the relative number of decays and termination locations for U and Th nuclei in each voxel. This allows for a direct calculation of the relative fraction of retained alpha particles, without using a surface area to volume ratio or any dimensional length measurements. This method is based on parent nuclide and crystal geometry data, so it is independent of any measured helium abundance or calculated age. It involves the

following steps:

1) Utilize the abundance of U and Th in each voxel of the 3D crystal volume to create the appropriate number of alpha particle daughter products. This is done by using the measured elemental abundance in the voxel as the basis for calculating the relative number of alpha decays each decay chain will generate over an arbitrary length of time.

2) Track the stopping location voxel of each alpha decay based on published average alpha stopping distances for each decay chain (Ketcham et al., 2011).

3) Build an accumulation model in a 3D volume representing the distribution of all alpha particles accumulated from U and Th decay within the crystal.

4) Calculate the fraction of total alpha decays retained within the crystal ($F_T$).


The Matlab script enacts these steps as follows. For a given voxel, the abundance of each of the three parent isotopes ($^{238}$U, $^{235}$U, $^{232}$Th) is used to generate a set of randomly oriented unit vectors for each of the alpha decays produced by parent nuclides in that voxel. Assuming secular equilibrium and using the ingrowth equation from Farley (2002) and the natural U isotopic ratio from (Hiess et al., 2012), the appropriate number of unit vectors per parent isotope per voxel is calculated as follows:

decays from $^{238}$U = 8 x [number of U atoms] x (1–1/137.818),

decays from $^{235}$U = 7 x [number of U atoms] x (1/137.818),

decays from $^{232}$Th = 6 x [number of Th atoms].

For each of the three parent isotopes, the randomly oriented unit vectors are multiplied by the appropriate mean alpha stopping distances. These are 18.81 μm for $^{238}$U, 21.80 μm for $^{235}$U, and 22.25 μm for $^{232}$Th (Ketcham et al., 2011). Each vector

represents a single alpha decay originating in the voxel. For a given voxel, this process creates the correct relative number of randomly oriented vectors of the appropriate length (alpha stopping distance) based on the parent isotope abundances in the voxel.

The next step is to model each alpha decay vector's stopping location. To do this we start by creating a 3D helium accumulation

array of zeros that is the same size as the tomography arrays. For each of the randomly oriented alpha decay vectors for a given voxel, the index location of the decay termination is determined by summing the (x, y, z) index location of the given voxel

(where the decay originated) with the (x, y, z) index values of the alpha decay vector. The vectors are randomly oriented and have non-integer magnitude (e.g. 18.81 µm for $^{238}$U decays), so (x, y, z) lengths of the alpha decays must be approximated to integer values to allow for tracking the voxel index location of the termination. We make this approximation by rounding each

index value to the nearest integer. This sum yields the index location of the termination of each alpha decay. That decay location is recorded by adding a value of 1 to that voxel index location in the helium accumulation array.

This process of modeling alpha decay vectors for each parent isotope and tracking the index voxel of the stopping locations is sequentially iterated through every voxel. The alpha stopping locations are tracked through the accumulation of alpha particles

in the 3D helium accumulation array. This process is carried out through a set of nested for loops in Matlab that iterate through every voxel in the 3D tomography arrays. Any voxels outside the crystal boundary have a zero value for U and Th, producing no alphas. The overall methodological framework is shown schematically in Figure 6.

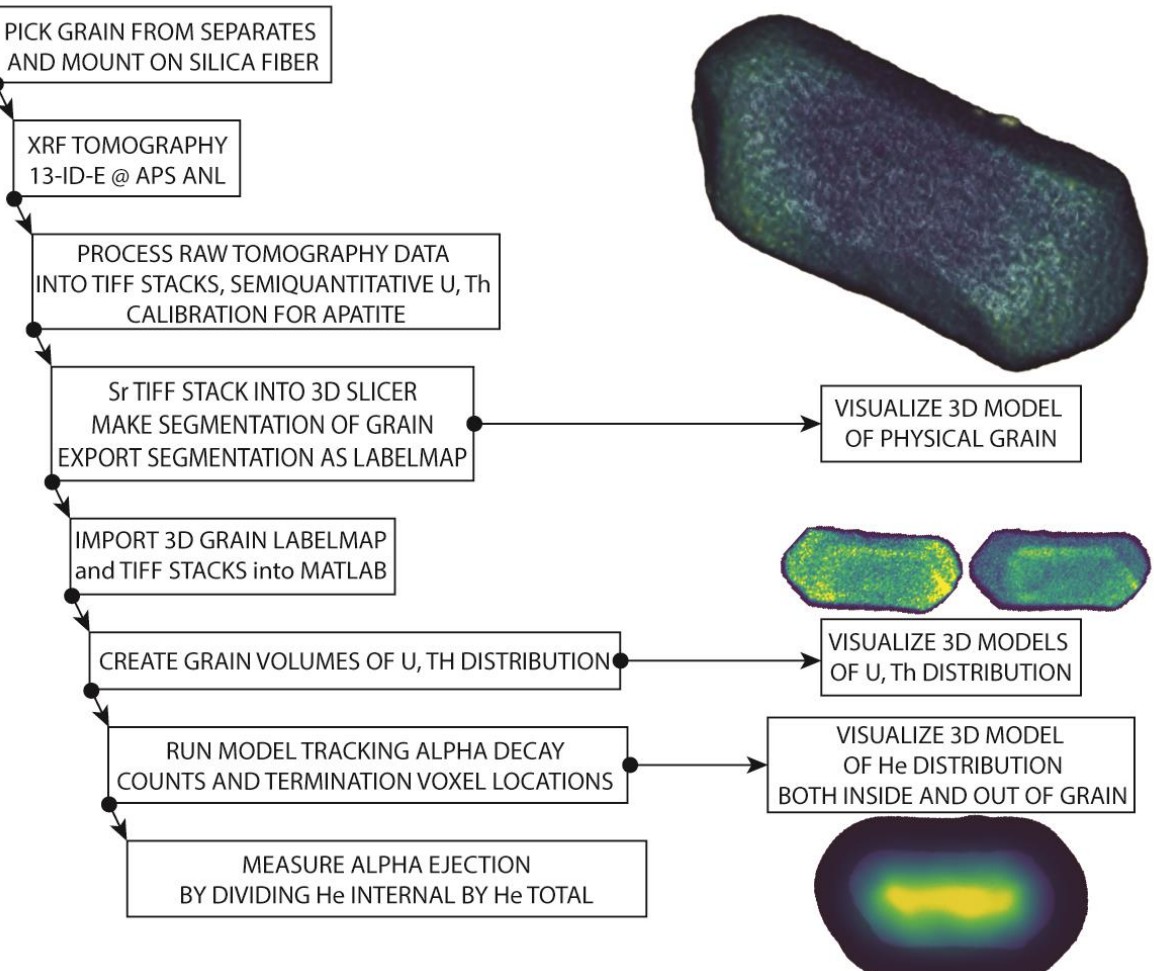

**Figure 6.** Schematic diagram showing methodological workflow described in the text. Visualizations are not to scale.

This process results in a 3D helium accumulation array that contains the daughter products of all alpha decays resulting from parent isotopes within the 3D crystal volume. Each alpha decay termination location is tracked spatially, including decay terminations outside the physical crystal dimensions (Figure 7). The dimensions of the 3D helium accumulation array is purposely set to be >30 µm bigger than the crystal in all dimensions, so it is large enough to capture ejected alpha particles. Because the helium accumulation array starts with zero values in every voxel, this excess emtpy space has no effect on the calculation. The fraction of total alpha decay terminations retained within the crystal ($F_T$) can then be calculated. To make this calculation, a helium retained 3D array is created by multiplying the helium accumulation array by the labelmap volume. $F_T$ is then calculated by dividing the sum of all accumulated alpha decay terminations located within the physical crystal boundary (helium retained array) by the sum of all alpha decay terminations, regardless of their location relative to the crystal boundary (helium accumulation array). For MGB5–2, the calculated $F_T$ value is 0.63.

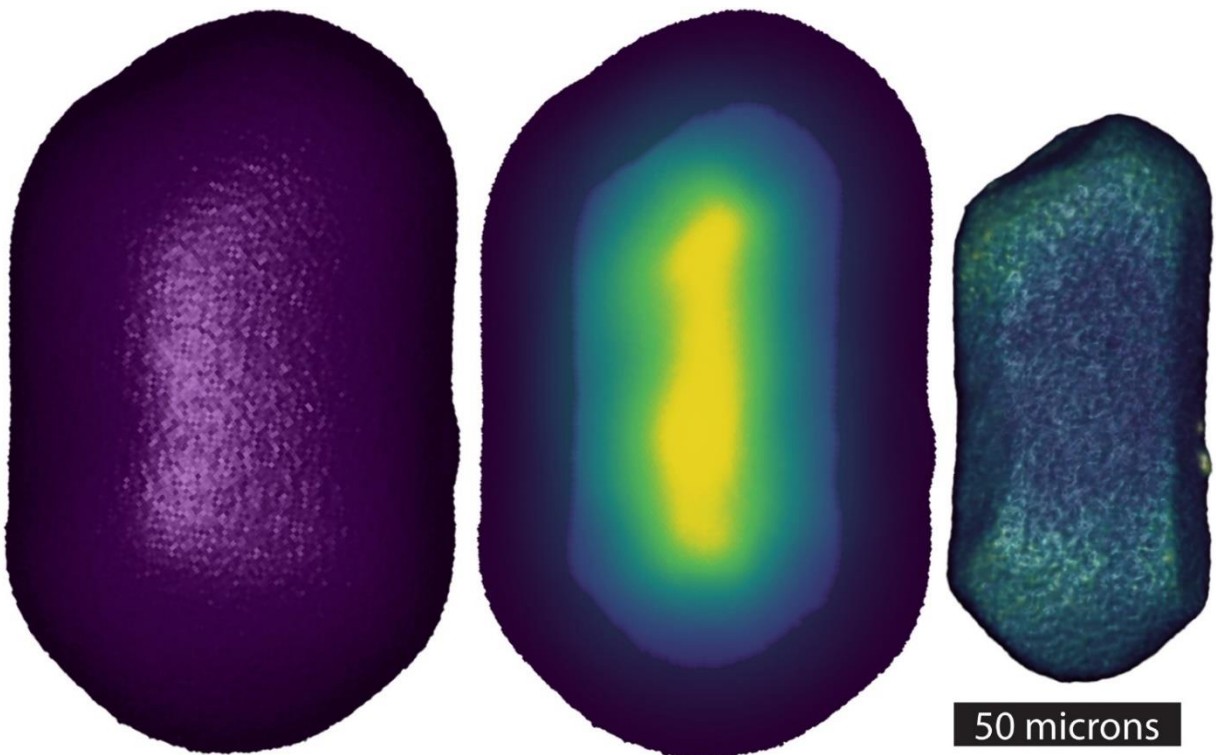

**Figure 7.** Visualizations of the 3D alpha decay tracking arrays. At left is the external surface of the helium accumulation array. The middle frame shows a partially transparent maximum intensity projection of the helium accumulation array (higher concentrations of He in warmer colors at the center of the crystal) as well as a partially transparent view of the physical crystal dimension overlain over the central portion of the crystal (faint blue internal outline). At right is the external surface of the 3D helium retained array, generated by multiplying the helium accumulation array by the labelmap volume.

**3 Discussion**

The method presented here of using synchrotron XRF microtomography to non-destructively measure the distribution of U

and Th throughout a crystal of apatite at a spatial resolution of 1 µm$^3$ represents a significant technological advance. The data resulting from this method can be used to develop a 3D model of alpha decay stopping locations and quantify the fraction of total alpha decays retained in the crystal for (U–Th)/He thermochronology. Given the importance of parent nuclide distribution for chronometry in Earth science, this new approach has many potential applications. In the following, we focus on applications to apatite thermochronology, but we recognize the breadth of potential implications and briefly discuss some of those as well.

**3.1 Alpha Ejection Correction**

Most published studies assume homogeneity of U and Th throughout a crystal and determine $F_T$ based on the expected alpha ejection ratio for a sphere with the same surface area to volume ratio as the crystal (after Farley et al. (1996)). Since its introduction, several updates and modifications to the original framework have improved this $F_T$ calculation. For example, Monte Carlo analysis can be used to estimate $F_T$ for a given geometry (Gautheron et al., 2012; Ketcham et al., 2011).

Additionally, both X–ray computed microtomography and 3D modeling from multiple optical images (Cooperdock et al., 2019; Glotzbach et al., 2019; Herman et al., 2007; Evans et al., 2008; Zeigler et al., 2023) have been shown to improve measurements of crystal geometry compared to estimates based on dimensional measurements. X–ray computed microtomography using a laboratory instrument (e.g., (Hanna and Ketcham, 2017)) shares many features with our method. It also allows for the non-destructive reconstruction of the 3D crystal geometry. The technique has a similar resolution to

synchrotron XRF microtomography and can reduce alpha ejection uncertainty due to crystal geometry. An important difference, however, is that laboratory instruments do not provide the trace element information we get from synchrotron XRF analysis, so they do not measure parent nuclide distribution and are less sensitive to the presence of mineral inclusions.

Attempts to incorporate U and Th zonation data into alpha ejection corrections are less well developed. Efforts to model the

impact of zonation on $F_T$ are mostly limited to no more than roughly 10 concentric zones and are often integrated into the calculation after converting from measured crystal geometry to spherical equivalent radius using surface area to volume ratio. Because of the low resolution and physically destructive nature of the quantitative zonation measurements in published studies, until now the common assumption of homogeneity or low-resolution zonation data has been appropriate. The resolution and non-destructive nature of synchrotron XRF tomographic data presented here make it possible to simultaneously 1) calculate

an $F_T$ value using a full crystal, high-resolution (~1 µm$^3$ scale) 3D model of U and Th distribution and physical crystal dimensions and 2) physically preserve the crystal for the standard (U–Th)/He analytical workflow of degassing to measure $^4$He and dissolution to measure parent nuclides by ICP-MS.

Given the potential for parent nuclide zonation to impact crystal ages on the order of 10–30% (Hourigan et al., 2005; Ault and

Flowers, 2012; Farley et al., 2011), it is possible that the commonly cited overdispersion amongst apatite (U–Th)/He single crystal ages could be ***fully remedied*** by accounting for parent nuclide zonation in $F_T$ calculation. For MGB5–2, the 3D $F_T$ value of 0.63 can be compared to calculated values using other methods to estimate the impact of this calculation on this crystal. First, we calculate the Th/U ratio by dividing the sum of the tomographically determined total number of measured Th atoms by the total number of measured U atoms, resulting in a Th/U ratio of 1.4, a close match to the multi crystal average

data from Mcaleer et al. (2009). Using this ratio, we then use two methods to estimate the homogeneous $F_T$ value for MGB5–2. The first is the Qt_Ft software package (Gautheron and Tassan-Got, 2010; Ketcham et al., 2011), where we input the simplified crystal geometry data: diameter of 60 µm, length of 120 µm, Th/U ratio of 1.4, regular hexagonal prism with 2 intact terminations. This method uses only geometric length data and Th/U ratio as inputs (not the detailed 3D crystal geometry or zonation data), and results in an $F_T$ value of 0.57. The second method we use to estimate the homogeneous $F_T$ value for

MGB5–2 utilizes the same Matlab script presented above, with the input tomographically determined U and Th abundance arrays replaced with homogeneous arrays with the Th/U ratio of 1.4. This method utilizes the 3D crystal geometry from the tomographic data but assumes parent nuclide homogeneity. This results in an $F_T$ value of 0.59. If we assume that the true $F_T$ value is the value calculated using the full 3D geometry and zonation data (0.63), we can calculate the percent deviation of the other two corrected ages from the corrected age using the true value. These deviations are 10.5% for the Qt_Ft value and 6.8%

for the value determined using the full 3D crystal geometry but assuming U and Th homogeneity (Table 2). We also show an example corrected age using each $F_T$ value for a 50 Ma uncorrected age for illustrative purposes.

| Method | $F_T$ | Example corrected age for 50 Ma uncorrected age | Deviation from 3D corrected age |
|---|---|---|---|
| Qt_Ft Synchrotron derived length and width measurement | 0.57 | 87.7 Ma | 10.5 % |
| Synchrotron 3D geometry and assuming homogeneity | 0.59 | 84.7 Ma | 6.8 % |
| Synchrotron 3D geometry and 3D zonation | 0.63 | 79.4 Ma | 0 % |

**Table 2.** Summary of calculated $F_T$ values for MGB5–2 using three different techniques described in the text, and the calculated age deviation based on each technique.

This result demonstrates how using high resolution zonation and geometry data to increase the accuracy of $F_T$ can significantly impact corrected (U–Th)/He ages (Hourigan et al., 2005; Farley et al., 2011). **We hypothesize that inaccurate $F_T$ values among apatite crystals from a single lithologic sample may be the primary driver of overdispersion in many apatite (U–Th)/He age populations, and that zonation differences amongst crystals control this inaccuracy.** If individual crystals from a single sample have unique parent nuclide zonation (different from each other), then the extent to which each age is

offset by zonation-controlled $F_T$ effects could vary widely in magnitude and sign. Future work could test this by applying the

technique presented here to several crystals from a sample with well documented individual crystal overdispersion. This would be followed by analyzing these same crystals using standard analytical techniques and calculating ages using multiple methods of $F_T$ estimation. The hypothesis leads to a prediction that the improved $F_T$ calculation may eliminate overdispersion in many single crystal age populations.


It is worth noting that in posing this hypothesis and focusing here on age variation driven by zonation-controlled $F_T$ inaccuracy, we do not discount the impact of helium diffusion kinetic effects on age variation. Rather, by shining a light on a newly measurable source of uncertainty in single crystal ages we hope to help work towards an improved understanding of how to best interpret age overdispersion broadly. Future work may help clarify which types of samples and cooling histories may be

most influenced by complex kinetic effects versus zonation.

## 3.2 Inclusions

The standard apatite (U–Th)/He analytical workflow relies on careful manual picking of suitable crystals for analysis using stereomicroscopy with cross-polarized light, often under alcohol immersion. This selection process tasks an individual to manually select crystals that do not host visually identifiable inclusions of other phases that might interfere with the

chronometer. In apatite crystals, such inclusions are sometimes zircon, monazite, apatite, or other phases. These inclusions can impact the analytical results through alteration of $F_T$ (due to different parent nuclide concentration in an inclusion compared to the remainder of the crystal), degassing profiles (different He diffusivity in an inclusion compared to the remainder of the crystal), and dissolution behavior during standard ICPMS parent nuclide measurements. Some laboratories partially overcome these impacts by using the more robust dissolution techniques common for zircon, but this does not fully account for the effect

that an inclusion can have on a single crystal age.

The synchrotron XRF microtomography method presented here provides a major improvement to the visualization and measurement of such inclusions. When an apatite crystal hosts even a very small zircon inclusion, it is easily identifiable in the synchrotron XRF microtomography results, notably due to decreased strontium, dramatically increased zirconium, and a

change in uranium and thorium in the inclusion compared to the surrounding area (Figure 8). Visualizing and quantifying the dimensions and parent nuclide abundance of such an inclusion could inform how to proceed with a given crystal. For example, an apatite crystal with a zircon inclusion could be discarded if other analyzed crystals lack inclusions. Calculating the effect of zircon inclusions on $F_T$ values for crystals fully imaged in high resolution would be a worthwhile future study. This would be helpful both for exploring the potential impact of inclusions on (U–Th)/He ages and providing a pathway to account for

that impact.

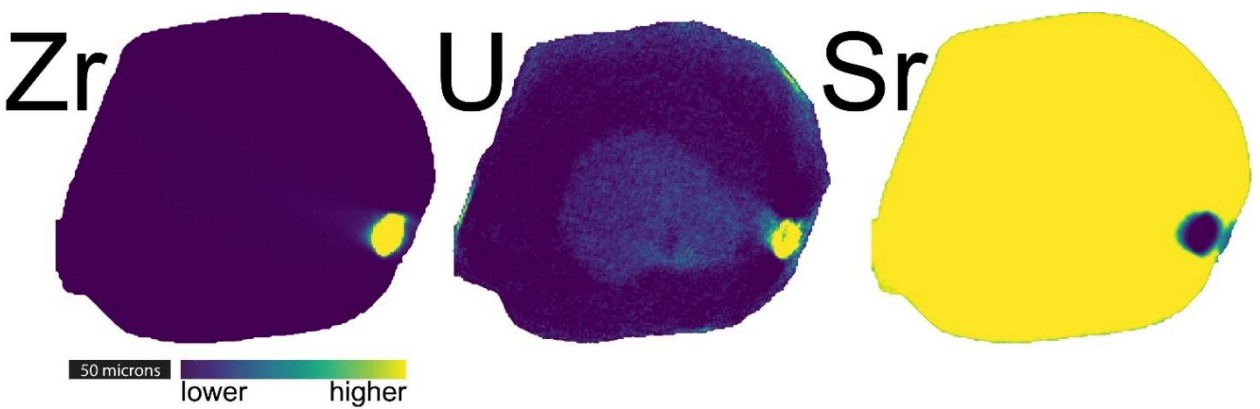

**Figure 8.** Three visualizations of a zircon inclusion in a single tomographic slice of an apatite crystal (AP1). The color scale is relative and designed to clearly distinguish between zircon inclusion and apatite crystal in Zr (left) and U (center), and Sr (right). This figure shows the
same virtual slice as that in Figure 2F.

When an apatite crystal hosts an apatite inclusion, it is identifiable in the 3D XRF microtomography data if it has elemental variation (particularly U and Th) at its boundary that can distinguish it from the remainder of the crystal. For (U–Th)/He thermochronology, this type of inclusion behaves similarly to any other type of zonation, and it is fully accounted for in the $F_T$
calculation method we presented earlier. Using traditional methods this type of apatite-in-apatite inclusion can be difficult to spot and may often be overlooked. Because the synchrotron XRF microtomography data allow us to accurately integrate the parent nuclides in these areas into the $F_T$ calculation, the determination of whether one of these areas is an inclusion or just an area of high uranium concentration in the crystal is unclear and unimportant. However, it is worth noting that in the MGB5–2 crystal several areas may be this type of inclusion, some of which are near the edge of the crystal and some of which are
internal to the crystal. While the sharpness of their edges varies, they do stand out as clearly hosting higher uranium than their surrounding areas. Because of the 3D distribution of these areas within the crystal, it is difficult to effectively visualize them in the type of 2D projection required for a figure. They are much more effectively visualized in movies included in the video supplement (Sousa et al., 2024a) and can be seen by downloading the 3D uranium volume file for MGB5–2 (Sousa et al., 2024b) and visualizing the data as a maximum intensity projection in 3DSlicer or other software packages.


In addition to the benefit to chronometry of measuring, identifying, visualizing, and quantifying inclusions, the synchrotron XRF microtomography methodology provides a new framework for studying zonation in unsectioned crystals generally. Our analytical sessions to date have focused on establishing a baseline set of data that demonstrably impact (U–Th)/He thermochronology, but along the way we have been able to visualize and quantify inclusions in apatite crystals. This method
has great promise as a tool for future studies of mineral and melt inclusions, and further work is needed.

### 3.3 $^4$He/$^3$He Thermochronology and Continuous Ramped Heating

The initial inspiration for this project emerged from the recognition that the $^4$He/$^3$He thermochronology workflow contains a critical decision point (Shuster and Farley, 2005; Shuster et al., 2005; Shuster and Farley, 2004; Sousa et al., 2016). After degassing a crystal, one must choose between two destructive techniques:

1) measuring the total parent nuclide abundances in the crystal of concern (by ICPMS) or

2) engaging in a study of parent nuclide zonation in the crystal of concern.

Until now, available analytical methods made these two options mutually exclusive. Whichever option was chosen, important data that could help build confidence in a $^4$He/$^3$He result was lacking. The synchrotron XRF microtomography method eliminates this dilemma. After XRF microtomography data are collected from a crystal, it is still physically intact for He

measurement and subsequent dissolution for measurement of total parent nuclide abundances.

Continuous ramped heating (CRH) is a newer technique that also probes the diffusion profile and diffusion kinetics of helium in apatite and other minerals (Idleman et al., 2018). Specifically, variations in the release curve measured using this technique provide information about features within the crystal structure that may host anomalous He or impose zones of anomalous

diffusivity. In addition to fluid, melt, and mineral inclusions, these include nanometer-scale voids or crystal defects. Zeitler et al. (2017) proposed that such sites might serve as helium diffusion sinks, an idea previously explored as an explanation for anomalous Ar diffusion by Watson and Cherniak (2003), and Guo et al. (2024) demonstrated that a model including diffusion sinks can explain anomalous He retention in apatite from a deep borehole. These proposed features are too small to be resolved using our tomography methods, but a more complete picture of U zonation and crystal geometry, coupled an accounting of

with the high-U fluid, melt, and mineral inclusions that we can resolve, will help to bolster these models and exclude confounding explanations for anomalous He diffusion.

The 1 μm$^3$ resolution model of parent nuclide abundance will allow for the development of a 3D diffusion model based on the measured crystal geometry and parent nuclide zonation. Because the current generation of interpretive tools for $^4$He/$^3$He

thermochronology and CRH data are rooted in low resolution models of crystal geometry and parent nuclide distribution (either assuming homogeneity or a small number of symmetric concentric zones), the application of this new type of data opens the potential for a new generation of interpretive tools that will improve our ability to translate $^4$He/$^3$He and CRH data into geologically meaningful thermal history constraints. The development of such tools is a focus of our ongoing efforts.

### 3.4 Implications for Interpreting Thermal Histories

The foundation of geologic interpretation of (U-Th)/He data is the translation from analytical data to thermal history information. This is often carried out using a heuristic or computational model. These span a wide range of complexity, ranging from basic interpretations of system averaged closure temperature (Dodson, 1973; Farley, 2002) to complex forward and

inverse models that use data from multiple samples in a geologic context and integrate calibrated diffusion models to constrain thermal histories (Ketcham, 2005; Gallagher, 2012; Sousa and Farley, 2020; Murray et al., 2022; Abbey et al., 2023; House et al., 1998). The most common proxy for tracking the influence of radiation damage on apatite diffusion kinetics is effective uranium (eU). This parameter is an aliquot average integrated parent isotope concentration, most often determined with an aliquot size of one crystal. It is directly integrated into the most widely used model for interpreting variations in diffusivity for apatite (U–Th)/He studies as eU = [U] + 0.235 x [Th] (RDAAM; (Flowers et al., 2009)). By integrating parent isotope concentration data on a crystal average scale, the RDAAM model implicitly assumes *homogeneity of radiation damage* throughout a crystal. By non-destructively imaging parent nuclide distribution, synchrotron XRF microtomography data provide a framework for future studies to explore the impact that zonation of radiation damage may have on helium diffusion in apatite. Heterogeneity of radiation damage in apatite crystals is a potentially important source of error in the application of models such as RDAAM to interpreting thermal histories. A more detailed future study of this seems worthwhile.

## 3.5 Other mineral phases

In addition to the advances in apatite thermochronology provided by whole crystal synchrotron 3D XRF microtomography, there is great potential for applying this method to other Earth materials. Perhaps the most obvious of these other phases relevant to chronometry is zircon, a primary target of high precision geochronology via the U/Pb system as well as the target for the zircon (U–Th)/He system. Analogous to the apatite (U–Th)/He system, the physical crystal geometry and the distribution of parent nuclides is fundamental to the geologic interpretation of zircon (U–Th)/He data, and XRF microtomography should have useful application in this realm. We have not attempted the technique on zircon crystals to date, but doing so seems worthwhile.

## 3.6 Methodologic availability

We have performed all analyses through a General User Proposal at the APS, a United States Department of Energy Office of Science User Facility. The APS is among the brightest synchrotron light sources, and in 2023-2024 is undergoing an upgrade to increase its brightness and capacity to engage in transformative scientific experiments. Because this methodology requires such a significant level of infrastructural support, another future direction of this project is to explore the potential to engage in synchrotron XRF microtomography of unsectioned crystals at other facilities. Additionally, it is of great interest to explore the potential for analyzing more crystals in a given amount of time by modifying tomographic settings to facilitate faster data acquisition (e.g. with lower spatial resolution). Because many of these facilities are heavily used by investigators across an extremely diverse swath of scientific endeavors, access to synchrotron facilities is limited for any given application, including XRF microtomography. Given the breadth of applications for this methodology across the Earth sciences, growth in this area is worth pursuing.

## 4 Conclusions

This work presents a methodology for measuring U and Th zonation in apatite through an application of synchrotron XRF microtomography at the APS. Tomographic imaging of the full 3D distribution of U and Th in a crystal of apatite at 1 $\mu m^3$ resolution solves the long-standing problem of documenting parent nuclide zonation non-destructively. This advance provides a pathway for eliminating the assumption of parent nuclide homogeneity in apatite that we hypothesize could be the dominant driver of overdispersion in (U-Th)/He ages. The data also facilitate a new generation of 3D models for (U–Th)/He and $^4He/^3He$ thermochronology. The application of synchrotron XRF microtomography to Earth materials has a broad set of future directions and potential applications to Earth science problems. We have discussed a few of these, particularly applied to geochronology and thermochronology, and we emphasize that the potential for this type of study is great and worth exploring further.

## 5 Appendices

**Appendix A: X-Ray Fluorescence Analysis**

XRF spectra were analyzed to give quantities (in ppm) for several elements seen in the experimental spectrum. This uses the known atomic levels and fluorescence yields for each element, so that each element is a sum of "hypermet" functions (giving a low-energy tail) for each of its known excited lines. For K-edges, this will include Ka1, Ka2, Kb1, etc with known relative probabilities. L emission lines are more complicated, but the values are tabulated.

The energy width of each peak is given by the Fano effect and a single "noise" parameter for the detector, $width = \sqrt{E \times Fano + noise^2}$, where E is the energy of the line, Fano a reduction factor (0.111 for Si). The elastic and Compton scattering are also modeled with a "hypermet" function, using independent "tail" and widths from the fluorescence peaks. Escape (the Si detector emitting Si Ka) is accounted for (see Ca and Fe in Figure A1). Energy calibration is varied in the fit. Pileup is accounted for as a scaled auto-correlation of the model spectrum with itself. Known detector thickness and path lengths of air and filters were used, but not varied in the fit.

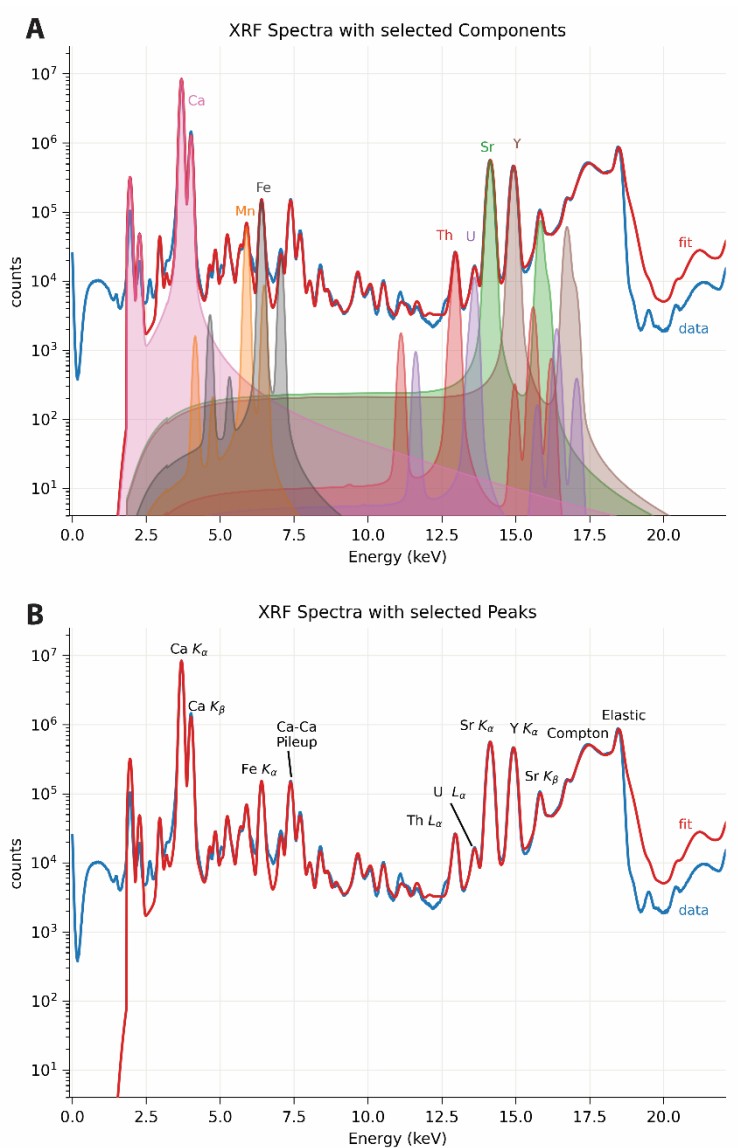

**Figure A1.** Measured XRF spectrum for all detectors and pixels from one slice of MGB5-2, before correcting for self-absorption. Fitted eigenfunctions for selected elements are shown in A. Selected peaks shown in B.


A representative fit is done to one (or a handful) of spectra, but not to the XRF spectra of each pixel. The fit for the spectra above used 26 Parameters: amplitudes for 14 elements, detector calibration (2), detector noise level, a pileup scale factor, an escape scale factor, and 7 parameters for the shape of elastic and Compton scattering to help account for the background. The fit gives an eigenfunction for each element (as shown in the figure) and an eigenvalue that is proportional to the abundance of

that element. A single scale factor can be found to convert all of these eigenvalues to mass per area or ppm levels. This is

similar in concept to the Dynamic Matrix analysis approach used in GeoPIXE (Ryan et al., 2015), but is implemented in the open-source Python package xraylarch.

With these eigenvectors, an XRF map for each sinogram (with ~180x140 individual spectra) can be decomposed using (fast) linear regression methods. This assumes that the response for any spectrum is linear in elemental abundance, which can be in error due to three effects:

1. pileup is taken to be a linear component in the linear regression of the spectrum.
2. peak width will increase with the count rate, especially as the detector gets close to saturation.
3. changes in the bulk composition of the matrix.

This linear decomposition approach is best for cases where spectra have fairly uniform total count rate and where the detector is not too close to saturation. We note that other XRF analysis methods will not do very well in these cases either.

These assumptions are pretty good for the samples and spectra analyzed here. We expect the quantification here to have absolute values better than order-of-magnitude level, as discussed in the main text. Abundance ratios between elements should be much better, and pixel-to-pixel variations in the distribution of each element even better.

**Appendix B:  Self-absorption correction for MGB5-2 apatite crystal**

We do not expect significant self-absorption for a 60-micron thick apatite grain for X-ray emission lines with energies greater than 10 keV.  Figure B1 shows the 1/e absorption length of apatite with selected element lines.  We would expect Ca to be
severely attenuated, but not Th, U, Sr, or Y.

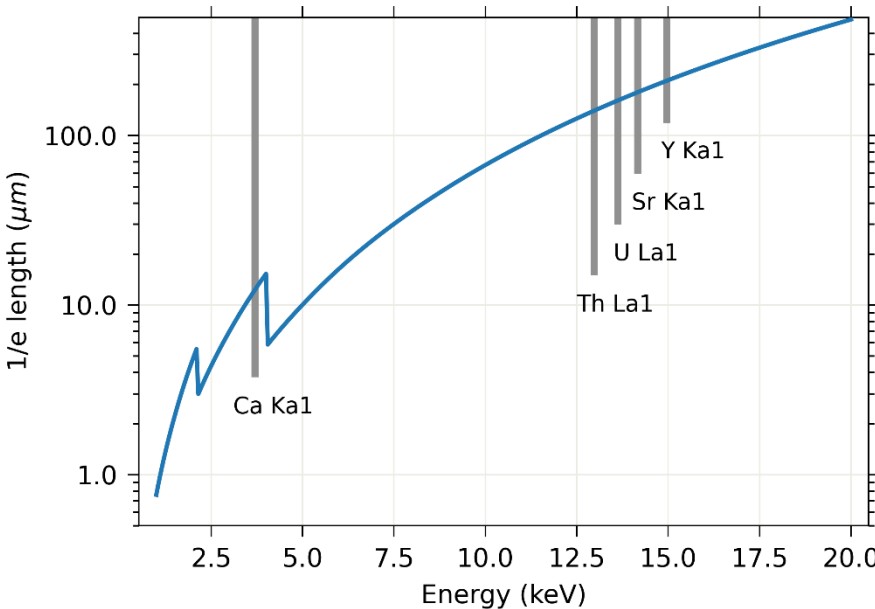

**Figure B1**.  Energy (keV) vs 1/e length (μm).  μ(E) is from the Python library xraydb.

A first-order correction was done assuming that absorption is dominated by the host material [$Ca_5(PO_3)_4F$, density=3.2 g/cm$^3$]
by multiplying the full energy spectrum for each pixel in each row (q value) by exp(mx), where x is the distance from the edge of the grain.  Self-absorption corrected data for MGB5-2 and python scripts used for this correction are published in Sousa et al. (2024b).  While some detector elements are out of the horizontal plane, they are about 60 mm away from the sample, so that the angle away from horizontal into those elements is very small.  The edges were identified for each q row using scikit-image processing tools.  Because the grain is ~60mm, the self-absorption effect is not very significant for the elements of
interest here (Th, U, Sr, Y).  In addition, because sinograms are collected over a full 360 degrees, each edge of the grain was "closest to detector" for one q scan. This reduces the effect of self-absorption by a factor of 2.  The following figures show the impact of self-absorption, particularly on calcium (Figure B2), and the results of our self-absorption correction (Figures B3 and B4)


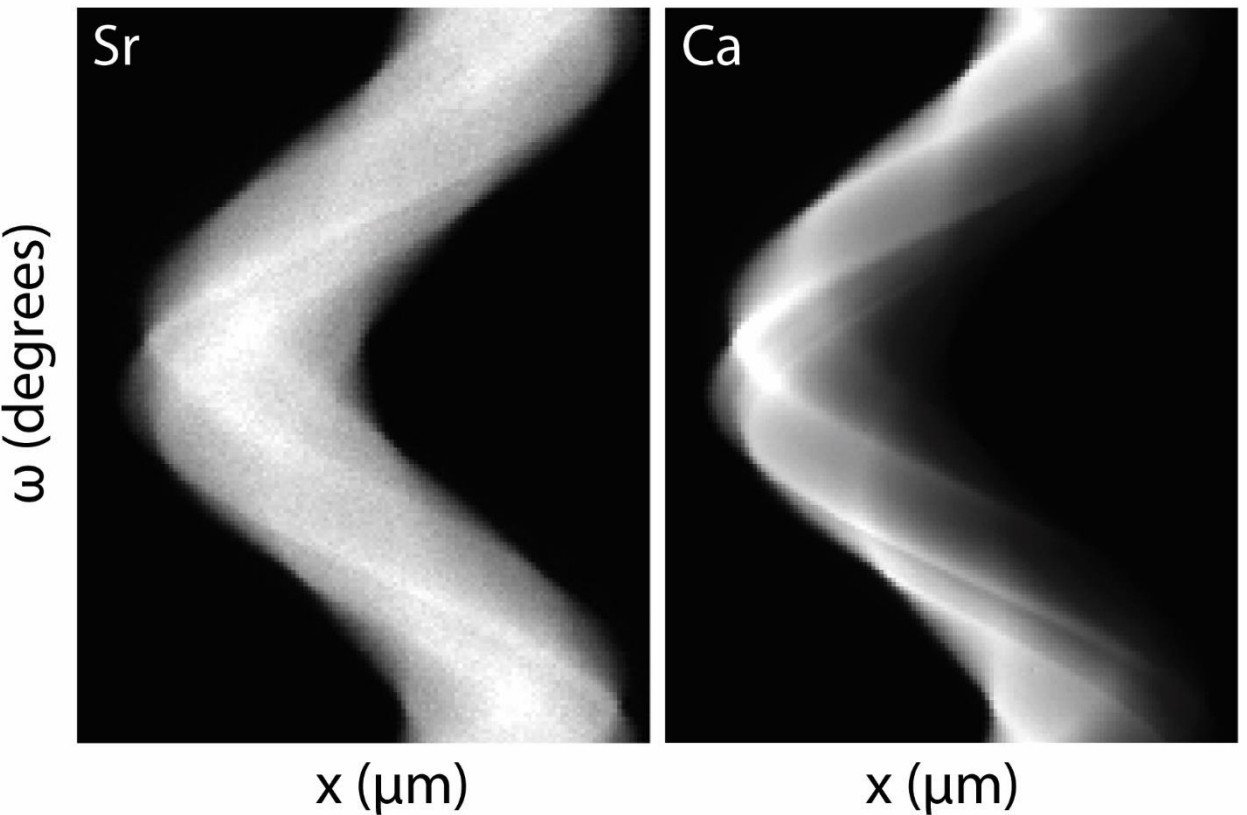

**Figure B2.** Sinograms of Sr and Ca from MGB5-2. The sinograms for Ca (right) show much higher self-absorption than those of Sr (left). Note that ω extends over 360 degrees.


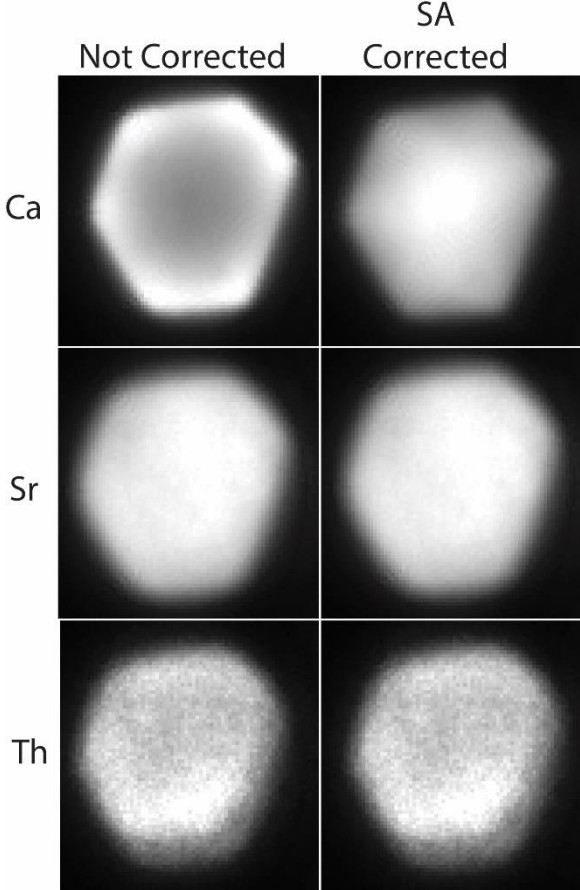

**Figure B3**.  The same virtual slice of MGB5-2 showing not corrected (left column) vs self absorption (SA) corrected result for Ca (top), Sr (middle), and Th (bottom).  As discussed in text, impact for Ca is significant, but minimal for other elements of interest.

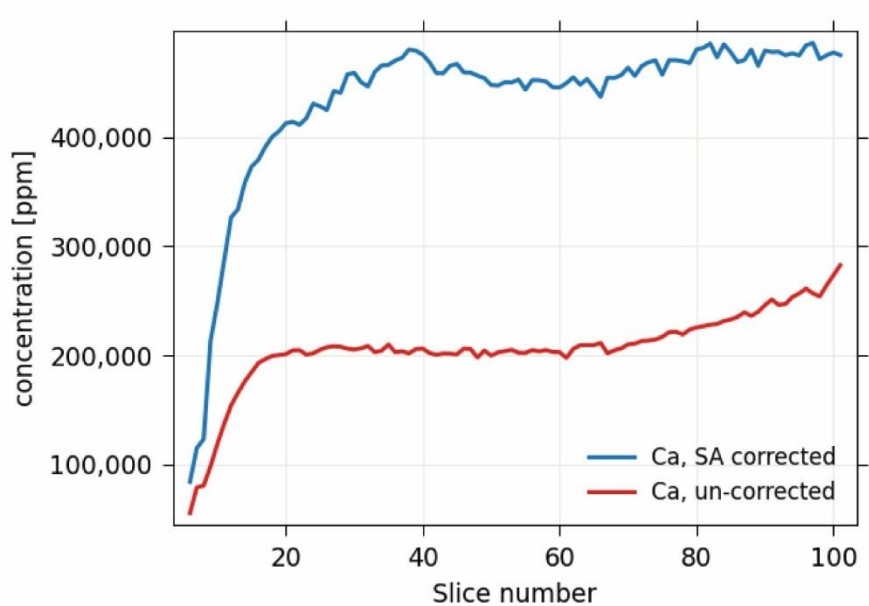

**Figure B4**. Plot of slice number versus elemental average abundance for Calcium, with un-corrected values shown in red and corrected values shown in blue. For apatite with stoichiometry $Ca_5(PO_3)_4F$, Ca is ~ 400,000 ppm.

**Code and data availability**

Data and code associated with this article are available in Sousa et al. (2024b). This includes synchrotron XRF data, elemental TIFF stacks for U, Th, Sr, and Y for crystals imaged in 3D, Matlab script, MGB5-2.ply 3D grain model, MGB5-2 3D volume files (U, Th, Sr, Y), self-absorption corrected TIFF stacks for MGB5-2, python scripts used for the self-absorption correction, as well as copies of the underlying files for the video supplement and abstract.

**Supplement**

Four supplemental videos and the video abstract are available in Sousa et al. (2024a).

**Author Contributions**

FJS, SEC, SRH, and ETR contributed to the conceptualization, sample selection, and analysis. AL and MN facilitated all aspects of synchrotron measurements and data processing of XRF data. FJS prepared the manuscript with contributions from all co-authors.

**Competing Interests**

The authors declare that they have no conflict of interest.

**Acknowledgements**

Portions of this work were performed at GeoSoilEnviroCARS (The University of Chicago, Sector 13), Advanced
Photon Source at Argonne National Laboratory. GeoSoilEnviroCARS is supported by the National Science Foundation – Earth
Sciences (EAR – 1634415). This research used resources of the Advanced Photon Source, a U.S. Department of Energy (DOE)
Office of Science User Facility operated for the DOE Office of Science by Argonne National Laboratory under Contract No.
DE–AC02–06CH11357. Access to sample AP (PRR–15064) is based on services provided by the Polar Rock Repository with
support from the National Science Foundation, under Award OPP–2137467. Special thanks to James Spotilla for providing
apatite separates from southeast Alaska.

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
