# Peer review of "U and Th zonation in apatite observed by synchrotron X-ray fluorescence tomography and implications for the (U-Th)/He system"

_Geochronology, 2024_

## Author Comment (AC1)

**REVIEW #1**

In the manuscript, titled "U and Th zonation in apatite observed by synchrotron X–ray fluorescence tomography and implications for the (U–Th)/He system", the authors describe a clear and well-documented novel method to investigate (U-Th)/He thermochronology. Great value is found in this manuscript, as the authors do a great job of making clear every step of the process they followed, in a way that allows the reader to reproduce it. Furthermore, the proposed method makes use of non-destructive analytical methods, which pose a great complementarity to the currently existing mainly destructive methods. The obtained results are fairly similar to the results obtained using the previous benchmark methods, and the authors discuss what and how these differences are caused.

All in all, this manuscript is very nice to read, the science within is sound and the story is well-supported by data. There are, however, some parts that could do with some additional elaboration. Please find my comments and suggestions below:

Abstract: in the first sentence, the "(…) in whole apatite crystals at the scale of one $\mu m^3$." is found to be misleading. Perhaps it is better to specify that this is the resolution scale, instead of the crystal size scale.

**We thank the reviewer for this clarifying comment, and will modify the text to improve clarity.**

P7, line 168: The authors state the focal spot size of the beam was $1\times2\mu m^2$ (V×H). Can the authors provide a measure of the beam's divergency, as this will affect the voxel size of the tomographic reconstruction, particularly at the edges of the sample.

**In the horizontal direction, the beam converges from 300 microns to 2 microns from a focusing mirror that is 200 mm away from the sample position. The beam will grow to 3 microns about 750 microns away from the ideal focal position. The vertical is a little better: although the size is 1.5 microns, the mirror is 400 mm away. So, the beam is collimated enough that the beam size will not double in size over the depth of the sample.**

P7, line 180: please provide the density of apatite that was used for these calculations.

**We used 3.2 g/cm3, and will add this detail to the text. See https://xraydb.xrayabsorption.org/atten/Ca5P3O12F/3.20/0.100/3000.0/21000.0/10 for an example.**

P10, line 207: "to determine an overall scaling factor between abundance for each element of interest and the full intensity of its fluorescence. These elemental scaling factors are then calibrated to give reasonable total abundances and used to produce sinograms of abundances": This statement is rather vague and unclear.

**We thank the reviewer for this clarifying comment, and we agree that adding more detail about XRF data processing would strengthen the manuscript. Rather than adding this method detail to the main text of this section, we suggest adding an appendix to the manuscript that would explain calculation of the elemental scaling factor in more detail.**

What do the authors mean by 'the abundance for each element of interest'? Is this the average abundance within the full apatite crystal, or some locally determined abundance? How did the authors determine either? This should be stated in the manuscript.

**We thank the reviewer for this clarifying comment, and add clarifying explanation to the text to explain that we use "abundance" to mean the concentration (values in ppm) for each element of interest. That is locally determined from the analysis XRF spectra that uses one calibration scaling parameter.**

In a similar way, it is unclear whether the approached method can indeed be followed to generate a sinogram of 'abundances', as of course every pixel in a sinogram in fact represents the total yield of a line through the sample (at a specific sample orientation).

**Yes, this is how computed tomography works. Every pixel in a sinogram has an intensity that is a line integral through the sample. By measuring multiple angles, the intensity for each pixel can be determined. As above, we converted sinograms of XRF spectra to sinograms of abundance. Using computed tomography methods, we then converted these to virtual slices and volumes of abundance.**

For instance, reconstructing said sinogram will not result in a virtual cross-section displaying the spatially resolved abundancies/concentrations (at least not without further scaling of said cross-section, using pre-existing data). Given how careful the authors are in describing the used methodologies, it would be opportune to elaborate on this (important) step as well.

**Reconstructions of sinograms do result in a virtual cross-section with spatially resolved abundancies/concentrations. As explained in the text, this does require a scaling factor from "weight of the elemental abundances" that is selected to give reasonable total abundances (in ppm) from the spectra of total grains. Again, we emphasize that these absolute values are best viewed as orders of magnitudes, while ratios of abundances of different elements will be more accurate, and variations of abundances for each element even more accurate.**

Later in the publication mention is made of ppm quantified results. Unless the above quoted sentence entails this, please provide (additional) information on how the quantification procedure was performed. In addition, a method validation (or reference thereof) would be greatly appreciated.

**Yes the previous quoted sentence does entail concentrations in ppm, as explained in the response to earlier comment from this reviewer.**

P10, line212: The authors state that XRF-CT attenuation correction is difficult and error prone. Although I agree with the former, and am sceptical about the latter statement, it may be opportune to cite here a few recent manuscripts that pertain this attenuation correcting, for readers who might be inclined to attempt to do so. Additionally, some methods exist that seemingly do not impose much errors in the final reconstruction, are fairly straightforward to use and do not require any chemical information input by making use of a neural network. These methods may be worthwhile to investigate.

**We thank the review for this comment, and we will change the language from 'difficult and error prone' to instead read "difficult to do with sufficient accuracy"**

**Because the absorption by these samples is dominated by the host apatite mineral, and not the trace metals, we could make a correction for each "X" line in each sinogram, assuming we can identify "the edge" of the apatite mineral, and assuming that the spatial heterogeneity of the trace elements does not significantly affect the self-absorption. When we have tried such processes, we find that it does not greatly improve the sinograms and does add some uncertainties.**

**The fluorescence energies for the elements of the interest are relatively high energy and have a fairly narrow range (ranging from 12.97 keV for Th La1 to 15.0 keV for Y Ka1), As shown, the attenuation by the sample will vary from 0 for X-rays emitted from the surface to a factor of 2 for X-rays emitted from the center of the sample. Thus, the fluorescence sinograms and the derived abundance**

**sinograms do not show significant self-absorption.  In contrast, sinograms for Ca Ka (3.7 keV), and even Fe Ka (6.4 keV) would be severely attenuated.**

**There are some published works on correcting self-absorption in X-ray fluorescence tomography, especially for lower-energy lines.  Since we do not use any such method, we do not cite these works.**

P 10, line 220: It could be noteworthy to mention that in principle U and Th signals can also be detected by their K-line excitation (in contrast to the L-line excitation used here), which corresponds to significantly higher energies (>90keV) and thus is virtually non-affected by attenuation effects in these samples. The drawback is of course that one has to obtain measurement time at high energy beamlines that allow microscopic focussing.

**The K-edge of U is 115 keV and its main emission line, U Ka is at 98.4 keV.  There are very few X-ray fluorescence beamlines that work at these energies.  The beamline used does not work above 30 keV.  The silicon detectors used in this study are more than 50% efficient for U La, but below 2% efficient for U Ka1 X-rays.  A Ge detector would be significantly better at 13% efficient.  If an X-ray energies above 115 keV was used to illuminate the sample, seeing Sr Ka and Y Ka would be very difficult, as these elements would have extremely low absorption cross-section.**

**It is true that self-absorption effects would be virtually non-existent for U Ka, these other effects would greatly compromise the data quality.  And, to re-emphasize: self-absorption in apatite crystals that are a few hundred microns in size is not a significant effect for U La1 or any of the other elements of interest in this study.**

P11, line 244: "Because strontium follows calcium and does not have an issue with self-absorption": It is more prudent to state that Sr has less of an issue with self-absorption, as also Sr characteristic fluorescence photons will be absorbed within the apatite matrix (~43% in 100µm apatite, per your calculations stated at p7, line 181). Additionally, would it not be possible to use Ca in any case, as the outer shell of the Ca should be visible/reconstructed and thus a total volume can be defined (assuming that there are no cavities within the crystal, perhaps this is what the authors fear/expect?).

**We thank the reviewer for this comment and will change the language to be clearer, stating that strontium is… "minimally impacted by self-absorption"**

**yes, Ca can be reconstructed.  It beautifully illustrates self-absorption, but since we do not use it in the analysis we have not shown that reconstruction.**

In any case, an intricate issue with such approaches is always where/how the difference between background and sample is determined. In the images provided in the manuscript, the background is cut away (white). Which threshold value did the authors set? How was this determined?

**The XRF fitting method does (try to) account for backgrounds in the XRF spectra, which comes largely from the tails of the elastic and Compton-scattering peaks. These scattering components are included in the set of eigenvectors used to linearly decompose the sinograms.**

**As a result, the sinograms of elemental abundances do fall off dramatically (to 1% of average intensity) within a few microns of the edges of the crystal.**

**The image reconstructtion algorithms used do automatically impose some constraints on the spatial extent of the data, but no additional constraint to spatially identify the edges of the crystal are imposed.  The intensities for the resulting virtual slices also drop off dramatically (again, to a few percent of the average intensity) within a few pixels of the edges of the crystal.**

**Several automatic algorithms were coupled to manual checking and modification procedures to delineate the physical grain edge in 3D slicer as explained in the text section 2.6.**

P12, line265: please explain the acronym SA/V equivalence

**We will modify the text here to not use an acronym for surface area to volume ratio.**

P13, line285: In the process, carefully described by the authors, usage is made of mean alpha stopping distances. At a later stage (line 295) these mean distances then essentially have to be rounded to the nearest integer value.

**This rounding does not have much of a biasing effect at all because a very small percent of total voxels are bisected by the segmentation boundary of the crystal. The voxels that are cut by the crystal edge can be expected to be roughly symmetrically distributed, resulting in an equal number of roundings in both**

**directions, and therefore not having much of an effect on the overall FT value. At lower spatial resolutions, this would become more of a problem.**

However, one could argue that using the mean alpha stopping distance is already an oversimplification of the physics that occur? Perhaps it would be more accurate to provide a sigma-deviation on this stopping distance, probed by another random number for each of the randomly oriented alpha decay vectors? Did the authors consider this, and is there a grounded reason why it was elected to forego this?

**In this case using mean stopping distances is not an oversimplification. The mean values are the averages of the well known stopping distances of each of the alpha decays in the three decay chaings (8 in U238 chain, 7 in U235 chain, and 6 in Th232 chain). These average values are the same used as standard practice for (U-Th)/He chronometry widely and is sufficiently accurate to correctly predict the FT values in this study.**

P14, Figure6: It would be interesting to provide the reader an estimate of the calculation time (e.g. s/CPU) required to perform these calculations. Additionally, the time required to complete these calculations will depend severely on the size of the investigated grains. Can the authors provide some insights on this size-dependency? Is this calculation time a bottleneck in the described method?

**The non-linear fit needed for an XRF spectrum can take 10s of seconds to a few minutes with a single processor of a typical modern workstation or laptop. While a single fit cannot be readily parallelized, fits can be parallelized across processors. As described above, we perform a few XRF fits per grain to check that the fitting model is well-behaved and the results sensible. The decomposition of XRF sinogram maps for a single slice through the sample from these fit results also take 10s of seconds. The iterative reconstruction methods take a few seconds per sinogram, which we did for each of the abundances found for each element of interest. Thus, the computation requirements are not too onerous, but many of these steps are best done initially with "partial supervision" using an interactive GUI application. All the needed analysis steps can also be fully scripted (using Python and the Larch package), so that the analysis for a grain can be set up and run in 10s of minutes to a few hours on a typical workstation or laptop.**

P14, line310: How do the authors explain that the outer dimensions of the (alpha decay) 3D array is >30μm bigger than the crystal in all dimensions, while the mean alpha stopping distance is (depending on the isotope) at most 22.25μm? Since the authors do not appear to use a probabilistic deviation on this stopping distance, one

could imagine that the alpha decay volume can maximally extend 22/23 (depending on rounding perhaps) μm beyond the outermost apatite grain voxels?

**The size of the alpha accumulation array needs to be sufficiently larger than the crystal dimensions so it has space to capture the ejected alphas from decays near the edge of the crystal. The array starts entirely full of zeros and accumulates alphas only in those voxels where decays terminate, so having it extra large has no effect on the calculation.**

P18, Figure8: is there a reason why the intensity scale maximum for the Sr image has been set to a relatively low threshold, thus rendering the entire image (at least apart from the Zr inclusion) yellow? A more appropriate scale may be selected, providing a more convincing (I expect noisy or heavily fluctuating in intensity?) image, that would still support the same conclusion i.e. that a Zr inclusion is present. Alternatively, a square root or log scale could be applied to the intensities for the Sr image, thus providing a more general overview of the collected data.

**We chose the color scale for Sr in figure 8 to maximize the visualization of the zircon inclusion in the virtual slice. For more clear visualization of Sr distribution in apatite see figure 5.**

General:

- The authors specify quite regularly throughout the manuscript that a 1μm$^3$ resolution/voxel size was attained. This clearly results in a good spatial resolution, but as seen from figure 2B also a 2μm$^2$ resolution provides quite clear spatially resolved information. This makes one wonder how the spatial resolution impacts the final results in terms of $F_T$ Would a 5μm resolution scan provide similar results, with the added advantage of a significantly higher sample throughput? A short paragraph discussing this would be of great value to the manuscript.

  **We are very interested in pursuing further experimentation to explore how the utlility of this application can be maximized by increasing throughput of samples. This does include lowering resolution, but doing so comes along with other complications and uncertainties, as discussed in the text (e.g. FT uncertainties, issues at grain edge, imaging small inclusions and highly complex zonation, etc.). We discuss this in section 3.6 and have added some text about tuning tomographic parameters to maximize utility, but at this point we have not completed enough experiments to know exactly how increasing to 5um resolution would impact results overall.**

---

## Author Comment (AC2)

**REVIEW #2**

"U and Th zonation in apatite observed by synchrotron X–ray fluorescence tomography and implications for the (U–Th)/He system" by Sousa et al. is an interesting and well-written manuscript that details a method to non-destructively visualize, in 3D, patterns of zonation in parent nuclides and zircon inclusions in apatite. The manuscript includes clear, step-by-step instructions with sufficient detail such that this study could be replicated. I believe this manuscript should be published in Geochronology and is valuable to the community. Below I outline some general comments followed by technical corrections and suggestions:

General comments:

The limited access to synchrotron facilities and beamtime seem to be an issue, as explained in Sect. 3.6. It would be helpful to have more details about how you got beamtime, how much it cost, and how long the actual analyses took.

**We agree that synchrotron access is important, and will add some detail about the proposal we used to gain access to the facility. For details about the length of time required for analyses, see text section 2.1 and 2.4.**

Could this method be modified or expanded upon for labs interested in acquiring data for multiple grains? For example, you have a grain imaged at a resolution of $2\mu m^2$ (Fig. 2 caption), does this significantly decrease the analytical time? Is it possible to mount several grains for analysis at the same time? I know you did not do this in this study, but it would be helpful to know if higher throughput is possible.

**We are certainly interested in finding a way to expand access to this type of data, decrease time required, etc, as discussed in section 3.6. Our thinking is very much in line with the type of questions listed here by reviewer #2, as we allude to in section 3.6. At the current stage, the tradeoffs between resolution, speed of analyses, and utility of information are not fully explored, but are of significant interest. We will add some text to section 3.6 better explaining this.**

I would like a little more justification for the samples you chose to analyze. Why analyze such small grains that have no demonstrated issues with unexplained overdispersion (that could have been attributed to zonation)? Did you attempt to analyze grains that have complex zonation?

**Section 2.2 discusses the criteria we used for choosing the samples measured to date. We agree with the reviewer that future work specifically studying**

**overdispersed crystals would be worthwhile, and we explicitly suggest this in Section 3.1 of the manuscript. The data shown in this manuscript includes a highly diverse set of types of complex zonation patterns, as shown in figures 1, 2, and 5.**

Are there (or could there be) issues with helium loss attributed to heating during analysis?

**We do not believe it is likely that x-ray illumination would significantly heat the sample. It will locally dump milliwatts of energy into a line through the sample that is 2um x 2 um extending the length of the sample. Since that power is spread over a line through the sample, it has ample opportunity to dissipate. Future work thermally imaging a crystal during analysis would explicitly answer this question. In the case that sample heating was sufficient to potentially drive He diffusion measurably, that could be fully ameliorated by degassing the grain and measuring He prior to XRF microtomography, rather than after.**

A table with sample names, (U-Th)/He ages and references, crystal dimensions, $F_T$ values, and the number of / a brief description of tomographic slices acquired and the resolution would be useful.

**We prefer to limit the figures and tables in the paper to those that directly contribute to understanding the new contribution provided by the synchrotron XRF microtomography method. We are explicit in the manuscript that although the thermal histories of these samples are of tectonic importance, they are not the focus of the paper. For these reasons we prefer to leave these details in textual form in the methods section.**

L97: Could you include photomicrographs of the analyzed grains? You state that you picked grains with no visible inclusions but in the video supplement for MGB5-2 there appear to be large near-surface inclusions.

**We do not have photomicrographs of all of the analyzed grains. The typical photomicrographs are taken primarily to document crystal dimensions, not presence or absence of inclusions, which require subjective interpretation by an individual, with use of crossed polars, ethanol immersion, and a very steady hand to visualize and interpret with certainty. The near surface inclusions visible in the MGB5-2 video are visible in the XRF data because of higher U and Th, but are also shown to be apatite in apatite inclusions as discussed in section 3.2, and would not necessarily be visible in photomicrographs.**

L175: Does the dot of epoxy covering the bottom termination of the apatite in the mount influence the signal?

**The epoxy will absorb some X-rays, which can be seen in the images, but it is not an important effect for these crystals, as it is relatively low density.**

L214-L220: You explain in Section 2.5 how concentrations were acquired and the uncertainties on them, but it is distracting that there is no ppm value attached to the color scale in Figure 2. I suggest referring the reader to the discussion in Sect 2.5 in the figure caption.

**We thank reviewer #2 for this suggestion, and will add the suggested reference to the figure 2 caption.**

L214-L220: Are there relevant citations for, or could you expand upon, how you arrived at the uncertainties on the concentrations? Are these grains degassed/dissolved yet? It would be nice to see if the "reasonable" abundances are actually in line with the actual abundances measured by ICP-MS.

**The concentration data shown are directly calculated from the XRF microtomography results, as explained in section 2.5.   The individual crystals imaged are neither degassed nor dissolved yet.  The absolute abundances from XRF are dominated by systematic errors from incomplete modeling of the solid angle and total efficiency of the XRF detector and will be factors of 2 to 5.  We encourage reading the values as order of magnitudes.  For MGB5 there are published aliquot average U and Th concentrations for the aliquots reported in Mcaleer (2009), the range of which is fairly consistent with the concentrations derived from the XRF tomography and shown in Figure 1 and 5.  However, we have not discussed this in the manuscript because we are explicitly not representing the XRF data as a useful absolute U and Th concentration measure (just order of magnitude).**

L240: In Table 1, for the definition of a labelmap, how are voxels that delineate the exterior of the grain classified?

**We explain the process of delineating the exterior of the grain immediately below Table 1 in the text of section 2.6.**

L245: I would like to see a brief discussion of what other elements (major? minor? trace?) can be visualized with this method. Subsequently, why did you choose to map Sr (and Y?) as a proxy for crystal volume rather than P?

**The XRF spectra and abundance analysis shows clear presence (and maps) for elemental maps for P, Ca, Mn, Fe, Sr, Y, La, Ce, Nd, W, Th, and U. There is some evidence of Cu and Zn.**

**While P, Ca, Mn, La, Ce, and Nd are visible, these are significantly attenuated (3 orders of magnitude) due to self-absorption by the sample and estimates of abundances would be very uncertain. We discuss this in the text of section 2.5 for Ca attenuation. Sr and Y both give very strong peaks, and are X-ray fluorescence energies that are both high enough to not be severely attenuated by self-absorption, and also near to the energies of Th, and U.**

**We have example XRF spectra with labelled peaks that can be included in an appendix if necessary.**

L297: You emphasize the value of the stopping distance several times throughout the manuscript but then round them. I understand why you have to do this. Do you expect that this has any appreciable impact on the calculated $F_T$ values? Do you expect the resolution of the mapped zonation (e.g., 1μm$^2$ vs. 2μm$^2$ vs. 10μm$^2$) to have an impact on the calculated $F_T$ value?

**We do not expect the rounding to have an appreciable impact on the calculated Ft values. Because of the high spatial resolution, the relative percent of total alphas that fall into a rounded voxel is low. Furthermore, since we round to the nearest integer, some will be rounded into the crystal, and some rounded out of the crystal, which would act to minimize any bias, so long as there is a roughly symmetrical distribution of percent of each rounded voxel inside vs outside the crystal. Decreasing the resolution of the measurements would certainly make this more of a problem, but the extent to which that would be an issue is not constrained here because we are only presenting data at the higher 1um3 resolution.**

L374: For illustrative purposes you calculate an $F_T$ value for a 50 Ma uncorrected age— this sample has a published mean helium date of 5.90 ± 0.42 Ma. Why the discrepancy? Additionally, the small size of the grain is going to have a large impact on the calculated $F_T$. Would it be possible to recalculate these data for a more typically sized grain?

**We explicitly use an example age that is highly deviated from the actual published ages to avoid any confusion that the example ages in table 2 are actually calculated ages (they are not calculated ages, just examples to show how the differences in Ft would lead to different corrected ages). The exciting new contribution here is the fact that we can directly calucate FT values based on the actual measured U Th distributions (not assuming anything about zonation).**

**Recalculating these data for a larger grain would require an assumption about zonation, and so we do not do so.**

L374: In Table 2, can the "Qt_Ft" method be additionally labeled as "traditional" or "geometric"? Does the Qt_Ft method use the Ketcham et al., 2011 equations for calculating $F_T$?

**We thank reviewer #2 for requesting clarification. We will add some text to the table to clarify that the QTFT method uses crystal dimensions derived from synchroton data (just length and width). QT FT does use Ketcham 2011 alpha stopping distances, as explained in the citations for QT FT.**

Technical corrections/suggestions:

- Add a line to the abstract that briefly describes the samples.

    **The samples are described in section 2.1, and the details of the specific samples do not bear on the fundamental contribution of this paper which is the focus of the abstract. Further, there is no reason to expect that this method wouldn't work on any apatites, and so including the details of the samples in the abstract would require clarification that goes beyond the brevity and focus we are trying to accomplish in the abstract.**

- Where there are multiple citations prefaced by 'e.g.,' or other words, there are extra parentheses (for example, L27, L29, L36, L50, etc.)

    **We thank the reviewer for this comment, and will make sure such typographical issues are fixed for publication.**

- L35: the listed stopping distances for apatite in Ketcham et al. (2011) are 5.93-22.25μm but you have 13-34μm listed?

    **We thank the reviewer for pointing this out. 13-34 um is the total range of all decay stopping distances for the three U and Th decay chains from Farley et al 1996, and 5.93-22.25 is the range of mean stopping distances for the three chains AND samarium from Ketcham. To avoid confusion we cite both papers and just list the rough average stopping distance of 20 um.**

- Adding a discrete color scale next to figure 2F would be helpful rather than explaining the color scale at the end of the figure caption.

**The three end member RGB map of 2F does not lend itself to visual color scale like the linear scale of the other frames, so we describe the three end members in the figure caption instead.**

- L222, L224: Gürsoy should have an umlaut.

- **We thank the reviewer for catching this, and will fix it.**

- L372: FT should be $F_T$.

**We thank the reviewer for catching this, and will fix it.**

- Consider citing Zeigler et al., 2023 (https://gchron.copernicus.org/articles/5/197/2023/) where discussing updates to $F_T$ calculations.

**We thank the reviewer for suggesting this citation, and will add it.**

---

## Author Comment (AC3)

**REVIEW #3**

"U and Th zonation in apatite observed by synchrotron X–ray fluorescence tomography and implications for the (U–Th)/He system" by Sousa et al.

I will preface this review by making it clear that my expertise lies not in geochronology but in experimental physics, in particular synchrotron-based data acquisition and analysis, with significant experience in XRF mapping. I therefore limit my review to the synchrotron methods and analysis.

This paper aims to increase the accuracy of (U-Th)/He thermochronology by accurately mapping U and Th zonation in apatite crystals using X-ray fluorescence tomography. The improvement in accuracy hinges on accurately measuring the U and Th concentrations throughout the apatite crystals. Table 2 suggests that this method increases accuracy over traditional methods by ca. 10%, and by ca. 7% over assuming homogenous elemental distributions.

However, I find a major flaw with this method:

"Importantly, this analysis does not account for attenuation of the X–ray fluorescence by the sample itself. This attenuation varies systematically with the energy of the main fluorescence line. Because the X–ray path through the sample to the detector varies for each x and w value, trying to account for this effect accurately would be difficult and error prone."

**We disagree with reviewer #3 regarding the interpretation that our treatment of attenuation by the sample is a major flaw with the study. In particular, this seems most likely related to the reviewers stated non-expertise in the details of the geochronology application, which is our primary focus. In particular, the biggest new contribution here is the high fidelity mapping of the relative distribution of U and Th spatially throughout a crystal. This is not significantly effected by self absorption. The uncertainties on absolute abundance estimates is openly discussed in the manuscript and does not have a negative impact on the geochronology application here.**

**Additionally, we have done a self absorption correction and have these results available to add to an appendix if necessary. Results show that the correction for Ca is significant but not overwhelming. The correction for the concentration of Th, U, Sr, and Y in the MGB5-2 grain are all minimal.**

The attenuation of the X–ray fluorescence by the sample itself, known as self-absorption, is significant. For example, for U fluorescence (16.366keV) the attenuation length (where 1/e of the initial intensity remains) is ca. 260 um, while for Th fluorescence (12.252keV) the attenuation length is ca. 130 um (https://henke.lbl.gov/optical_constants/atten2.html). For sample MGB5–2 (diameter < 100um) this may not be an issue, however, for 03PH307A- 2 and AP-1 with diameters more than 200um this will be significant and cannot be ignored.

**The X-ray path for 03PH307A- 2 was about 60 microns.   The crystal was oriented so that the shorter axis was the path taken by the X-rays to the detector.**

**As the reviewer may know, "attenuation length" means the intensity has dropped by 1/e (to 37% of the initial intensity).   The attenuation length for Th La1 (which is 12.97 keV) is 140 microns.  The attenuation length for U La1 is 160 microns.   Th La1 X-rays from the far side of the grain (60 microns) will be attenuated to 66% of the initial intensity.  U La1 X-rays from the far side of the grain will be attenuated to 69% of the original intensity.    That is the worst case here: the average depth is only 30 microns (so 80% and 83% of the initial intensity for Th and U, respectively).   One can conclude that XRF analysis without correcting for self-absorption will result in Th/U ratios that are systematically high by up to 5%.**

**We are claiming that absolute abundances are "better than order-of-magnitude" levels but could easily be off by factors of 2.  The relative abundances (say the Th-U ratio) should be much better than that, though we are not claiming that they are correct to 2 significant digits.**

**The concerns of the reviewer here do not have a significant implication for the primary results of this study (the 3D imaging of U and Th relative abundance in apatite crystals).  It seems most likely that the reviewer is mistakenly interpreting the large uncertainties on absolute abundance measurements (which we openly and explicitly address in the manuscript) as a flaw in the study, which it is not.**

The error introduced from not accounting for self-absorption would be significant, and likely exceed the claimed benefits. Indeed, correcting for this is difficult, but solutions to this problem exist, for example:

1. Zichao Wendy Di, Si Chen, Young Pyo Hong, Chris Jacobsen, Sven Leyffer, and Stefan M. Wild, "Joint reconstruction of x-ray fluorescence and transmission tomography," Opt. Express25, 13107-13124 (2017)

2. Yang, Q., Deng, B., Du, G., Xie, H., Zhou, G., Xiao, T. and Xu, H., Q. Yang*et al.*. X-Ray Spectrom., 43: 278-285 (2014)

3. Gao, J. Aelterman, B. Laforce, L. Van Hoorebeke, L. Vincze and M. Boone, "Self-Absorption Correction in X-Ray Fluorescence- Computed Tomography With Deep Convolutional Neural Network," inIEEE Transactions on Nuclear Science, vol. 68, no. 6, pp. 1194-1206, (2021).

Additional comments.

1. How long does each tomography slice take to acquire? How many could you collect in 1 day of beamtime?

**This would vary significantly depending on crystal size and choice of tomography parameters (corresponding to spatial resolution).  For the data presented in this paper, we collected roughly 1 full crystal of tomography data per 1 day of beam time.  We agree that this is useful information to include and will add it to the manuscript. With upgrades to X-ray facilities and detection electronics, the beamline will probably be able to go between 2 to 5 times faster without degrading data quality.**

2. Is there an application to a lab source? What would need to happen to be able to apply this to a lab?

**If this refers to collecting such data using a laboratory X-ray source, that seems unlikely. This work used a micro-focused X-ray beam with very high flux:  the focus beam size will determine the ultimate pixel size and the flux was on the scale of $10^{12}$ monochromatic photons/sec.   It would probably be feasible on a micro-focused bending magnet beamline with 1000x lower brightness, but probably be considerably slower to achieve similar data quality or have a much lower count rate that would make U distribution a bit more difficult to see in a reasonable amount of time.**

3. Using self-absorption corrections, what is the viable upper limit of sample size?

**This would depend on the density and elements to be analyzed.  For apatites, 200 microns would certainly be correctable, and 500 microns might be possible – Ca might be uncorrectable, but Th, U, Sr, Y probably would be correctable.   For biological samples (density below 2, mostly water and carbon), samples as large as 1 mm are feasible.   We note, however, that data collection for a given spatial resolution would go as the square of the size of the object. That is, a grain 200 microns across would take 10x as many data points to get the same spatial resolution as a grain that is 60 microns across.**

4. Can the homogenous approximation be estimated with a simple 2D fluorescence scan? In this case the is there an upper size limit to the measurement? This seems like a possible way to achieve a measurable improvement with significantly fewer experimental complexities.

**It is unclear what reviewer #3 is referring to by "homogeneous approximation" here.   One can do 2-D XRF mapping.  For grains such as those analyzed here, quantification of XRF spectra to get abundance would be complicated by the non-uniform sample depth.  Making a thin section to give a uniform thickness (say, 30 microns) would be the standard analytic approach in that case.**

---

## Author Response (AR2)

Dear Frank and coauthors,

Thank you for revising your submission based on the feedback from the three reviewers. I think these revisions have made your manuscript a stronger contribution. I have some minor points that I think you will be able to address quickly, after which I anticipate accepting your manuscript for publication.

FT inaccuracies as the dominant source of overdispersion: The revised version of the manuscript emphasizes/doubles down on your hypothesis that FT inaccuracies due to parent nuclide zonation are the dominant driver of age overdispersion (e.g., bolded text at line 389, added lines 498-499 in the discussion). As you point out in section 3.4 however, zonation will also lead to nonhomogeneous radiation damage within crystals, which could lead to complex helium diffusion kinetics and also produce age dispersion. I'm wondering if you can provide some rationale as to why you think FT corrections will be dominate over such kinetics effects? I suspect that the answer might require some nuance; e.g., radiation damage-induced complex diffusion kinetics might be much more important for old samples with protracted cooling histories, whereas FT inaccuracies might have a greater influence on rapidly cooled late Cenozoic samples.

**Thanks for pointing this out. We also think the appropriate response is fairly nuanced here. Most importantly, we are explicitly NOT trying to make an argument for the lack of importance of kinetic effects on age dispersion. Rather, we are trying to interpret this studies new data to its logical conclusion. By this we mean that the data clearly demonstrates that zonation driven Ft inaccuracy is another (non kinetic), potentially large magnitude uncertainty source that we have never been able to measure before. And we now can measure it. Because we do not intend to make an argument against the importance of kinetic effects, we would rather not introduce new text making that argument. So to clarify our thinking on this topic we have added a new short paragraph to the discussion shortly after the bolded hypothesis around line 389 that explains our thinking here.**

You also bring up implications for 4He/3He thermochronology in section 3.3.; I think it would be valuable to also mention here what role the parent nuclide mapping could have in the continuous ramped heating (CRH) approach that the Lehigh group has developed, which in my opinion is quite complementary to and more accessible than

4He/3He. At least some of the published CRH datasets document helium release characteristics that are complex but unlikely to be explained by parent nuclide zonation alone. For example (and I realize this paper literally just came out), Guo et al. (2024) document that samples from depths in the KTB borehole at which all helium should be diffusively lost in fact contain significant helium and have nonzero, overdispersed (U-Th)/He ages. This result cannot be explained by zonation-induced FT inaccuracies.

**Thank you for this helpful comment. We have added some text regarding CRH along with the section about 4/3, as well a few citations.**

Line 509: is there a word missing after 'known'? Or should excited lines be removed from the parentheses?

**Thanks for pointing this out, the parenthetical needed to be removed. We fixed this.**

Line 511: It's not clear why you transition to a bulleted list here – is there a sentence missing before the list?

**It works for us to remove the bullets and write out the lines in paragraph form.**

Figure A1: It's not clear in some cases which labels go with which plotted components (e.g., Ca pileup, Y, and Compton/Elastic). Can you relabel for clarity?

**We added a second frame in this figure so that we could both show the selected elemental components (A), as well as a more clearly labelled version without components shown (B).**

Thank you for taking the time to carry out these last edits, I look forward to receiving your revised manuscript.

Marissa Tremblay
Purdue University
June 24, 2024

Guo, H., Zeitler, P.K. and Idleman, B.D., 2024. Behavior of helium diffusion sinks in apatite: Evidence from continuous ramped heating analysis of borehole and wellcharacterized samples. Earth and Planetary Science Letters, 641, p.118828. https://doi.org/10.1016/j.epsl.2024.118828